# Neuronal diversity and stereotypy at multiple scales through whole brain morphometry

Yufeng Liu [1], Shengdian Jiang [1,2,20], Yingxin Li [1,3,20], Sujun Zhao[1,3], Zhixi Yun [1,2], Zuo-Han Zhao [1], Lingli Zhang[1,3], Gaoyu Wang[1], Xin Chen[1], Linus Manubens-Gil [1], Yuning Hang[1], Qiaobo Gong[1], Yuanyuan Li[4], Penghao Qian[1], Lei Qu [1,4], Marta Garcia-Forn[5,6,7,8,9], Wei Wang[6,7], Silvia De Rubeis [5,6,7,8,9], Zhuhao Wu [10,11,12], Pavel Osten[13], Hui Gong [14], Michael Hawrylycz [15], Partha Mitra [13], Hongwei Dong [16], Qingming Luo [17,18], Giorgio A. Ascoli [19], Hongkui Zeng [15], Lijuan Liu [1,3] ✉ & Hanchuan Peng [1] ✉

We conducted a large-scale whole-brain morphometry study by analyzing 3.7 peta-voxels of mouse brain images at the single-cell resolution, producing one of the largest multi-morphometry databases of mammalian brains to date. We registered 204 mouse brains of three major imaging modalities to the Allen Common Coordinate Framework (CCF) atlas, annotated 182,497 neuronal cell bodies, modeled 15,441 dendritic microenvironments, characterized the full morphology of 1876 neurons along with their axonal motifs, and detected 2.63 million axonal varicosities that indicate potential synaptic sites. Our analyzed six levels of information related to neuronal populations, dendritic micro-environments, single-cell full morphology, dendritic and axonal arborization, axonal varicosities, and sub-neuronal structural motifs, along with a quantification of the diversity and stereotypy of patterns at each level. This integrative study provides key anatomical descriptions of neurons and their types across a multiple scales and features, contributing a substantial resource for understanding neuronal diversity in mammalian brains.

Neurons are the fundamental units of the nervous system, and their morphological analysis is crucial to understand neural circuits[1]. One salient feature of mammalian neurons is their extensive, long-range axonal projections across brain regions[2]. However, our understanding of neuronal morphology and function is limited by the incomplete digital representation of neuron patterns[3,4]. Recent studies have focused on full neuronal morphology, including both dendrites and axons, using genetic and viral techniques that label neurons sparsely[5–8]. In combination of these labeling techniques, various imaging modalities, such as serial two-photon tomography (STPT)[9], light-sheet fluorescence microscopy (LSFM)[10,11] and fluorescence

micro-optical sectioning tomography (fMOST)[12,13], have been employed. These techniques have produced an extensive volume of imaging data, primarily hosted by the Brain Research through Advancing Innovative Neurotechnologies (BRAIN) Initiative - Cell Census Network (BICCN) community[14].

Recent studies emphasize the importance and advances of generating complete neuron morphology reconstructions, particularly long projecting axons[15–18]. However, analyses of the complex arborization patterns of axons in mammalian brains are still limited. Analysis of the dendritic arborization has also been limited to traditionally defined morphological features, but is largely missing the overlay with

brain anatomy to yield rich spatial information. Additionally, there has been little work on integrating information from neuronal populations, individual neurons, and sub-neuronal structures[19].

In our effort to analyze morphological patterns of neurons at different scales, we consider the statistical distributions that quantify both the diversity and stereotypy of morphological patterns[16,20]. Across different "types" or "classes" of morphological patterns, a diversity metric describes the variety among different types of morphological patterns and their respective degrees, while a stereotypy metric quantifies the level of conservation of patterns within each type. Neurons may differ greatly in their morphological, physiological and molecular attributes[2,21–23]. Despite previous efforts to study the diversity and stereotype of various neuron types, such as hippocampal interneurons[24], striatal neurons[25], and cortical neurons[16], a systematic analysis at a whole brain level and across multiple scales is still missing.

Our study makes an initial effort in describing the diversity of conserved morphological patterns of neurons at various anatomical and spatial scales in the context of whole mouse brains. Using a massive number of light-microscopic images of mouse brains generated by the community of Brain Research through Advancing Innovative Neurotechnologies (BRAIN) Initiative - Cell Census Network (BICCN), we performed an analysis of 3.7 peta-voxels of images with which we also reconstructed thousands of annotated neurons, and developed one of the largest available multi-morphometry datasets. By analyzing patterns of neurons at six structural scales, we discovered conserved morphological modules (collections of brain anatomical regions) and motifs (sub-neuronal structures) distributed throughout entire brain. This effort allows us to characterize the mouse brain anatomy based on a detailed, multi-scale description of neuron morphologies. Furthermore, we also attempted to establish a model explaining how features of different scales have complementary effects on morphological characterization. By combining the diversity and stereotypy scores at different scales, we visualized and identified the modularized structure of brain regions or neuron types, which were defined based on soma location, projection, or lamination information, at single-neuron resolution.

## Results

### Brain mapping of multi-morphometry data generated from peta-voxels of neuron images

Our collaborative effort with BICCN and partners has led to the assembly of one of the largest collections of single-neuron morphology data in mice. This 3.7 peta-voxels dataset included 204 whole-brain images captured at micrometer and sub-micrometer resolutions using fMOST, STPT, and LSFM techniques (Fig. 1A; and the meta information including transgenic lines, main targeted neuronal types, and many other information are summarized in Supplementary Data 1). We call this image dataset IMG204 to simplify the subsequent description. We analyzed these images captured by different modalities to investigate the modular organization of brains and associated patterns across anatomical scales. To facilitate an objective comparison of morphological patterns across different imaging modalities and experimental conditions, we registered all IMG204 images to the Allen Common Coordinate Framework version 3 (CCFv3) atlas[26], using the cross-modality registration tool mBrainAligner[27,28] (Fig. 1A, Methods). Indeed, the sparsely labeled populations of neurons in different brains could be accurately aligned to study the colocalization relationship of their patterns (Fig. 1A).

To demonstrate the utility of our data analysis framework, we produced quantitative descriptors of patterns at various morphological scales, from the entire brain to the resolution of individual synapses. To do so, we developed a cloud-based Collaborative Augmented Reconstruction (CAR) platform[29], which is a software package with multiple computational tools for high-throughput generation of multi-morphometry. We performed semi-automatic annotation of a total of 182,497 neuronal somas from 122 fMOST brains (Fig. 1B; Supplementary Data 1; Supplementary Data 2) using an initial automatic soma detection, followed by collaborative annotation through a mobile application CAR-mobile, available on the CAR platform. We call this soma dataset SEU-S182K, including detailed information of brain ID, soma-location in 3-D, and registered brain region (Supplementary Data 3). As neurons were often labeled with different degrees of sparsity in these brains, we captured the large variation of soma distribution across various brain samples. We achieved this by annotating both brains with very sparsely labeled neurons and also brains with densely labeled neurons. Overall, in 72% (88/122) of the brains in SEU-S182K, there are more than 100 annotated somas. Spatially, among 314 non-fiber-tract regions in CCFv3 (CCF-R314; Methods), 296 regions contain annotated soma (Fig. 1B). For specific brain regions, such as the caudoputamen (CP) and the main olfactory bulb (MOB), We identified over 20,000 somas and high densities of up to 1710 and 2576 somas/mm$^3$ respectively.

We then traced both the dendritic and axonal morphologies of individual neurons with annotated somas. For dendrites, we constructed a database, called SEU-D15K, which contains 15,441 automatically reconstructed 3D dendritic morphologies. We cross-validated the brain-wide reconstructions in SEU-D15K with the dendrites of 1876 manually curated neurons and found similar distributions of morphological features (Supplementary Fig. S1A; Methods) and Topological Morphology Descriptor (TMD) scores (Supplementary Fig. S1B). Overall, SEU-D15K dendrites showed consistent morphological features, with exemplar tracings for various projection subtypes aligning well with visualized neurite signals (Supplementary Fig. S14). We observed distinct morphologies for representative tracings for the brain stem, cerebellum, forebrain, and neuromodulatory centers (Supplementary Fig. S15). However, dendrites with somas in proximity, particularly those in the same brain regions, usually clustered closely (Supplementary Fig. S1C). To derive a spatially tuned dendritic feature vector with high discrimination power, here we extended our recent spatial tensor analysis of dendrites for human neurons[30] to analyze these mouse dendrites in SEU-D15K. Subsequently, we developed a dendritic microenvironment representation to characterize local neighborhood information around a target dendrite (Fig. 1C; Methods). Due to the higher precision of location information available in mouse brains compared to human surgical tissues[30], we were able to construct the dendritic microenvironment could also be constructed to describe the spatially tuned dendrite structures (Methods). In this way, we produced 15,441 dendritic microenvironments corresponding to SEU-D15K and used this approach to quantify the dendritic diversity and stereotypy as described below.

Using our framework of multiscale morphometry (Fig. 1D; Methods) that spans resolution levels from centimeters to micrometers, we analyzed morphological patterns of labeled neurites from 191 mouse brains that containing detectable neurites on low resolutions (Fig. 1A, D), as well as accordingly generated dendritic microenvironments (Fig. 1C) and fully reconstructed neuron morphologies. We extended our analysis to the SEU-A1876 dataset, which contains fully traced 3-D morphologies of 1876 neurons, including their complete dendrites, proximal axonal arbors, and distal axon arbors (Fig. 1D). The dataset primarily consists of projection neurons, mainly located in the Thalamus (37.2%), Cortex (24.1%), and CP (16.8%), with 702, 455, and 317 neurons, respectively. We specifically extracted 3776 densely branching axonal arbors, 1876 dendritic arbors, as well as the primary projection tracts connecting such arbors (Fig. 1D). We also identified axonal bundle motifs as diverging, converging or parallel projecting patterns. Additionally, we detected 2.63 million axonal varicosities from the axonal arbors, and accordingly pinpointed the respective synaptic patterns (Fig. 1D).

Our analytics framework covers six major scales of morphological patterns (Fig. 1D): Neuronal populations, dendritic microenvironments,

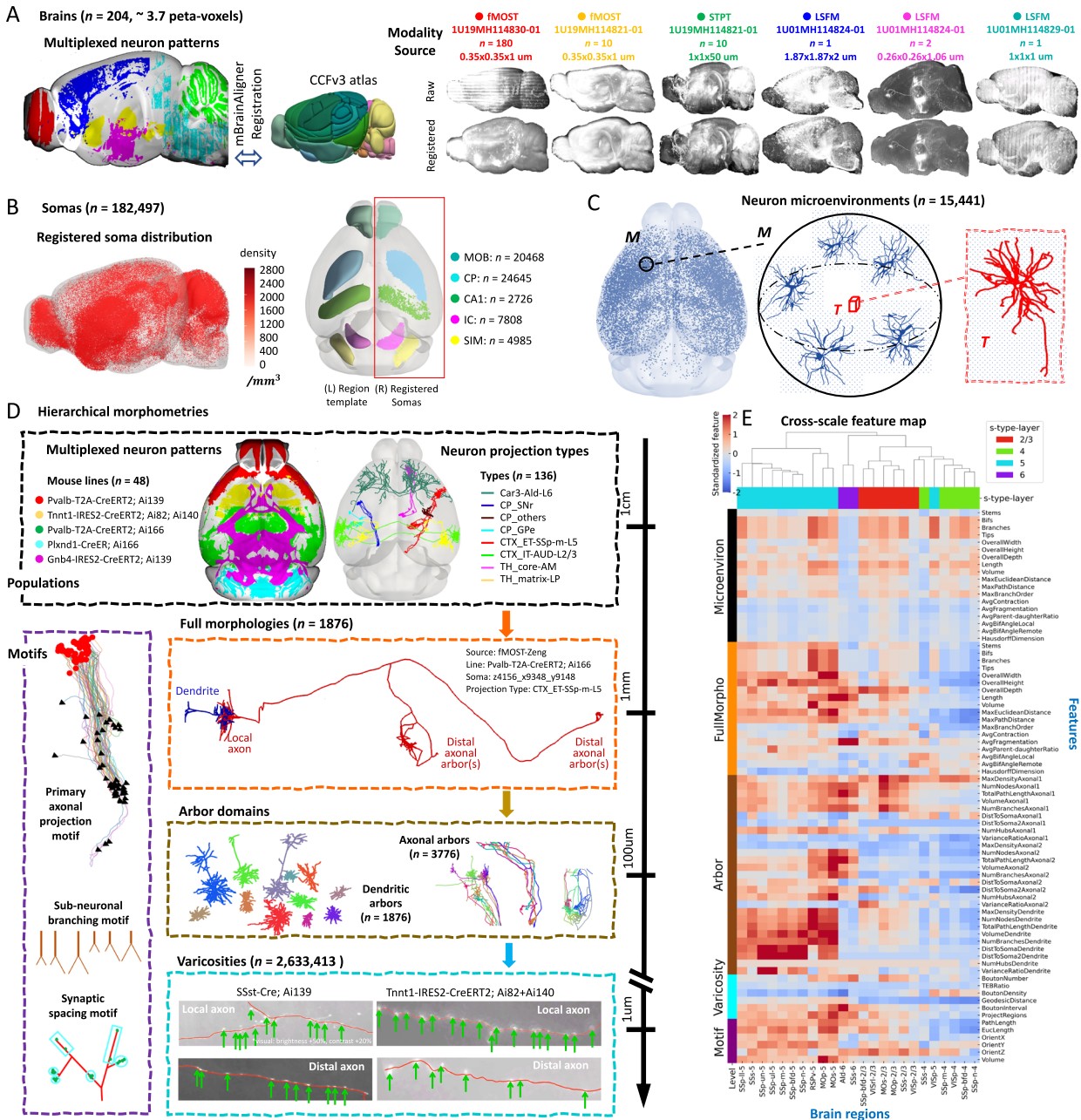

**Fig. 1 | Multiscale morphometry analysis from mouse brain images. A** The mouse brain dataset IMG204 comprises 204 brains (3.7 peta-voxels) of 3 different modalities (fMOST, STPT, and LSFM) obtained from 4 BICCN projects. Left, a multiplexing view displays salient voxels on the sagittal middle sections of six mouse brains from different sources. The salient voxels are colored by image sources. Middle, the CCFv3 atlas that all brains are registered to. Right, representative sagittal maximum intensity projections of whole-brain images from each modality and source. Imaging modality, research group, the number of brains collected, and typical voxel size are specified at the top. **B** Left, sagittal view of the spatial distribution of 182,497 semi-automatically annotated somas on the CCFv3 template, along with their densities (color bar). Each soma is represented by an individual dot. Right, horizontal projection of five regions (color-coded) along the anterior-posterior (AP) axis (left) and respective soma locations as dots (right).

**C** Left, horizontal projection of auto-traced dendritic morphologies (SEU-D15K). Middle, dendritic microenvironment (M) representation for each neuron (target). A microenvironment is a spatially tuned average (see Methods) of the most topologically similar neurons (up to six neurons, including the target neuron) within a distance of 249 μm from the target neuron. Right, morphology of the target neuron within the microenvironment on the left. **D** Multiscale morphometry. Hierarchical representation including representative visualizations for six scales of morphometries ranging from centimeters to micrometers, i.e., neuron population (mouse lines and projection types), full morphology, arbor, motif, varicosity, and the microenvironment displayed in (**C**). **E** Heatmap of the cross-scale feature map for lamination subtypes of cortical neurons (s-type-layer). Soma types (s-types) with their soma located in the same cortical lamination were grouped together. Source data are provided as a Source Data file.

single-cell full morphology, sub-neuronal dendritic and axonal arborization, structural motifs, detected axonal varicosities, along with quantitative characterizations of the diversity and stereotypy of patterns at each level. We quantified a number of morphological features to characterize properties of brain regions as well as individual neurons whenever possible (Fig. 1E). Cross-scale feature maps demonstrate high

potential for cell typing and subtyping, with anatomically similar regions generally exhibiting analogous morphology throughout the whole brain (Supplementary Fig. S2A). Moreover, lamination and projection patterns emerge as prominent factors in grouping subtypes of cortical neurons, based on cross-scale features (Fig. 1E, Supplementary Fig. S2). Our analyses also revealed that broadly distributed yet highly discriminating

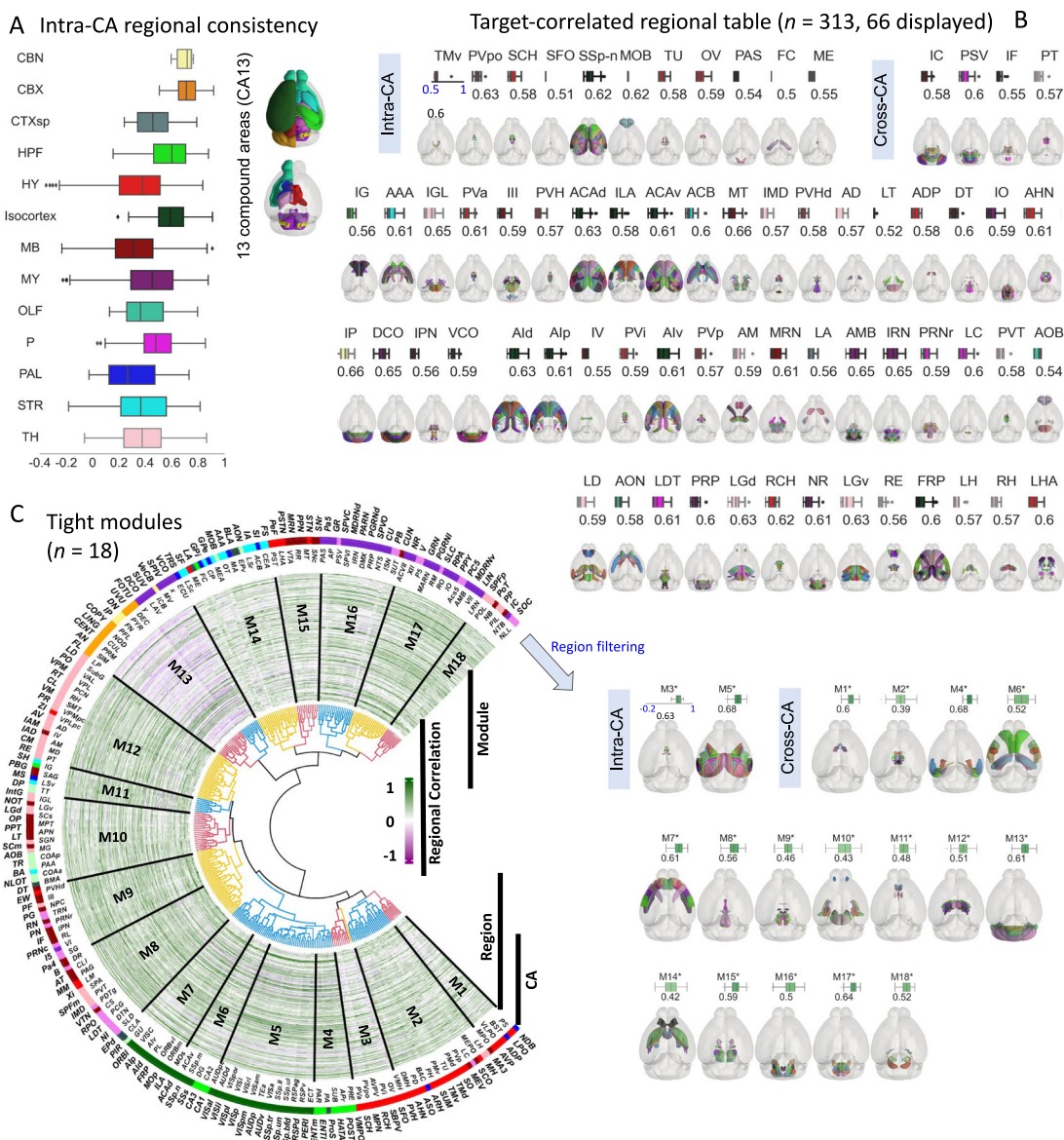

**Fig. 2 | Modules inferred from multiplexed brains. A** Intra-Compound Area (intra-CA) consistency. Left, box plot of the intra-CA consistencies for 13 compound areas in the brain (color-coded). Right, the 13 compound areas in CCFv3 atlas. A compound area is a super-region composed of functionally correlated CCFv3 regions. Sample size: CBN, 4; CBX, 14; CTXsp, 7; HPF, 15; HY, 44; Isocortex, 43; MB, 39; MY, 44; OLF, 11; P, 26; PAL, 9; STR, 14; TH, 44. **B** Horizontal projections on the CCFv3 template of regions with a Spearman correlation coefficient of at least 0.5 with the target region (specified at the top of each image). Each image is accompanied by a box plot that shows the distribution of the pairwise correlations between these regions and the target region, with the box colored by CA as in (**A**). Region sets are categorized as intra-CA if all regions are within the same compound area, and cross-

CA if they span across at least two compound areas. The first 66 region sets (out of 313) are displayed. **C** Whole-brain co-occurrence modules. Left, circular heatmap representing the neurite density distribution for each CCFv3 brain region (*N* = 314) as radial 191-element vectors (number of brain images). The dendrogram shows how brain regions cluster to form modules. Labels for each region are specified on two outer layers of the graph, with corresponding compound areas are labeled in the colored circle. Right, tightly inter-correlated modules, with modular consistencies (pairwise Spearman correlations) shown in the box plots on the top of the brains. Source data are provided as a Source Data file. Box plot: edges, 25th and 75th percentiles; central line, the median; whiskers, 1.5× the interquartile range of the edges; dots, outliers.

---

features across multiple scales could be integrated (Supplementary Fig. S2). We have outlined the key novelties of our approach and related findings pertaining to the six morphological scales (Supplementary Data 8).

**Inferring brain modules using multiplexed brains**

For morphological patterns visible in the range of millimeters to centimeters, we analyzed the diversity and stereotypy of neuron populations labeled in IMG204 (Fig. 1D). Quantifying the conservation or reproducibility of morphological patterns (stereotypy), in functionally established anatomical regions helps define whether these patterns

are sufficiently consistent to make biological inferences. On the other hand, capturing the diversity of these patterns not only confirms expected variations across brain regions, but also validates the accuracy in aligning images during brain multiplexing.

We developed an algorithm to segment the neurites in IMG204 (Methods), and used the co-occurrence of these neurites over the entire set of image samples to infer the diversity and stereotypy of the respective neuron populations. We grouped all 314 brain regions defined in CCFv3 into 13 larger regions (compound areas, CAs). Each CA corresponds to sets of functionally related brain regions within the CCFv3 taxonomy (Fig. 2A). We found that several CAs, such as

isocortex, cerebellar cortex (CBX) and cerebellar nuclei (CBN), have more tightly correlated intra-areal neurite patterns compared to other CAs (Fig. 2A). Within each CA, the positively correlated neuron populations (Fig. 2A) imply covarying brain patterns in IMG204.

We sought to identify highly correlated brain regions for each of the 314 CCFv3 regions ("target"), resulting in the discovery of 313 sets of individual regions that exhibit a Spearman correlation no less than 0.5 with their respective target regions (Supplementary Fig. S13; Supplementary Data 4). For each of these sets, we identified one or more matching brain regions whose neurite patterns correlate most strongly with the patterns in the target (Fig. 2B, Supplementary Data 4). 11 sets involve regions in the same CAs (intra-CA), while the remaining 302 involve regions from different CAs (cross-CA). Regions in most of these 313 sets, however, turn out to be immediate neighbors that share region borders (Fig. 2B; Supplementary Fig. S13). Examples include the pair of caudoputamen (CP) and globus pallidus – external segment (GPe) for which we previously reported single neuron level projection[16]. These results suggest that stereotyped "connections" of neurites exhibit a noteworthy degree of consistency with the brain anatomy delineated in existing brain atlases like CCFv3.

The observation above motivated us to further search for modules of brain regions that share similar co-occurring neurite-patterns as tight clusters (Fig. 2C). We identified 18 non-overlapping, intercorrelated, tight modules from the hierarchical dendrogram (Methods). 16 of which are cross-CA, composed of neighboring regions from multiple compound areas (Fig. 2C, Supplementary Fig. 13, Supplementary Data 5, Methods), which highlight hubs of co-occurring neurites. For example, the M3* module includes the medial preoptic nucleus (MPN) that closely associated with various regions, including the anterior, intermediate, and preoptic parts of the periventricular hypothalamic nucleus (PVa, PVi, PVpo)[31]. M5* encompasses four auditory cortical regions and five somatosensory regions, suggesting possible associations between auditory and somatosensory functions in mice[32]. The M7* module contains regions linked to the primary motor area (MOp) circuit, either as input (gustatory area, GU; dorsal part of the agranular insular area, AId) or output (GU; AId; ventral and posterior parts of the agranular insular area, AIv, AIp; orbital areas, ORBm, ORBl, ORBvl)[33]. In the M8* module, regions such as the dorsal tegmental nucleus (DTN), laterodorsal tegmental nucleus (LDT), and raphe regions like dorsal nucleus raphe (DR) are known to play roles in the regulation of sleep and circadian rhythms[34,35]. In the M10* module, we found the intergeniculate leaflet of the lateral geniculate complex (IGL), dorsal and ventral parts of the lateral geniculate complex (LGd, LGv), olivary pretectal nucleus (OP), superior colliculus (SCm, SCs), nucleus of the optic tract (NOT), and anterior pretectal nucleus (APN). These regions are part of the projection circuit from thalamic GABAergic neurons involved in circadian responses to light[36]. In the M14* module, regions are either involved in the basal ganglia circuits, including CP, GPe, and GPi, or they are part of the projection from the amygdaloid to CP[37]. The majority of regions in the M15* module are associated with glutamatergic and GABAergic regulation[38].

### Discovering brain parcellation using dendritic microenvironments

We used the diversity and stereotypy of single neuron morphological patterns to further delineate brain modules. We first examined the dendritic patterns of individual neurons within SEU-D15K (Fig. 1C). In this dataset, each local dendrite was reconstructed within a soma-centered cuboid approximately 28.52 million $\mu m^3$ in volume (Methods), -57 times of the larger volume compared to a recent study delineating local dendrites in cortical L4 neurons[39]. Our dendritic reconstructions are distributed in the majority of CCFv3 regions (222/314). To characterize the neuronal architecture in local neighborhoods, we extracted a 24-dimensional feature vector for each dendritic microenvironment. This vector aggregated both the dendritic

morphology of individual neurons and the spatial relationship of neurons in a small neighborhood (Methods). Next, we used a minimum-Redundancy-Maximum-Relevance (mRMR) algorithm[40] to select the top three discriminating features. We mapped those to the CCFv3 atlas to produce a 3-D brain-wide RGB-coded microenvironment map, where each channel corresponds to one feature (Fig. 3A). This RGB-coded representation facilitates the visualization of spatial variability in microenvironment features across the entire brain.

Whether dendritic features can be leveraged to distinguish cell types is debated[41,42]. However, without complete and accurate dendrite reconstructions we are limited in these efforts. Unfortunately, existing labeling techniques pose challenges to reconstruct error-free entire dendrite arborization. For pyramidal neurons, reconstructing precisely both basal and apical parts of dendrites is difficult, as apical dendrites can extend substantially. Neuron partition methods such as G-Cut[43] cannot avoid loss of information, either. In our dendritic microenvironment approach, we mitigated these problems by prioritizing accuracy over completeness, focusing on precisely reconstructed local dendrites surrounding somas to improve classification.

One remarkable observation is that despite the limitations of the approach, the microenvironment map shows clear boundaries that align with the primary CCFv3 region borders (Fig. 3A). For example, CP neurons are clearly distinct from cortical neurons. Cortical layers can also be discriminated based on these features, adding on observations from conventional soma-density method[26,44], axon projections[16], or a full description of the apical-basal dendrites of cortical neurons. Indeed, while each of the three color-coding features has a different distribution (Supplementary Fig. S4D–F), they jointly define a number of anatomical details that are consistent with the CCF parcellation.

Based on the diversity of brain regions indicated by the dendritic microenvironments, we identified six major clusters of regions (Fig. 3B). In the shown example, most laminated cortical neurons share similar feature patterns, placing them in one of the major clusters, although they could be further clustered hierarchically. Hippocampal neurons in CA1 and CA3 are clustered away from cortical, striatal, and thalamic neurons (Fig. 3B). Indeed, the hippocampal neurons have similar average straightness and Hausdorff dimensions like most other cortical neurons but differ in variance percentages (Supplementary Fig. S4D–F). CP neurons, instead, have a distinct pattern compared to other striatal neurons (Fig. 3A, B).

Within each microenvironment cluster, however, neurons show clear stereotypy. To measure the conservation and transition of these features within or across brain regions, we took an approach guided by the definition of four axial projection paths (Fig. 3C). The first path follows the tangential direction along the lamination of cortical layers. Cortical neurons share relatively stable features until entering the entorhinal area, lateral part, layer 5 (ENTl5) (Fig. 3D – Path1). The second path, orthogonal to the first one, clearly reveals reduced "variance-percentage" and "straightness" when entering and leaving CP (Fig. 3D – Path2). The third path following inner side of the border of CP and nearby regions shows the different distributions for the three features, which means that local dendrites along this path have strong heterogeneity. Thus, along the third path, there is a high likelihood that a variety of cell types can be encountered (Fig. 3D – Path 3). The fourth path, from the inner side to the outer side of CP, indicates a linear gradient of the "variance-percentage" and "straightness" features for CP, albeit with opposite trends (left and middle of Fig. 3E).

As the CCF anatomy was essentially developed using expert-annotation of an averaged brain of registered individual brain-images to determine the boundary of anatomical regions[26], we also generated the CCF-space average of all 191 brains to measure the visible contrast of previously defined brain regions or subregions compared to what we could observe using the microenvironment approach (Supplementary Fig. S16). The microenvironment features are able to identify more variation across and within brain regions compared to the

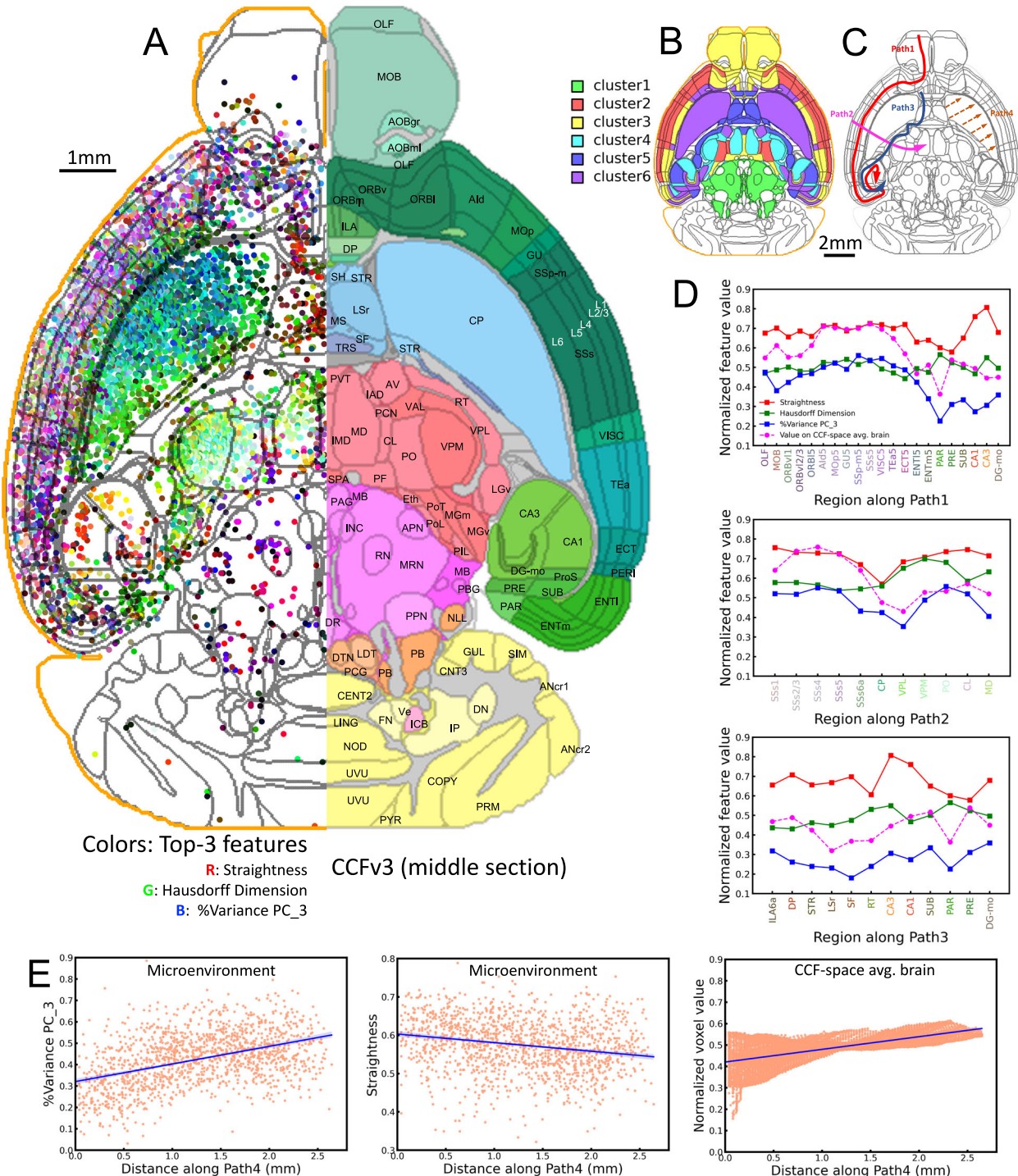

**Fig. 3 | Feature distributions of dendritic microenvironments across the whole brain. A** Left, the three most discriminating features of microenvironments —average straightness, Hausdorff-dimension, and variance percentage of PC_3 are visually represented as colored points on the middle axial section of the CCFv3 atlas. Right hemispheric microenvironments were flipped to the left hemisphere. The outer boundary of the CCFv3 template is indicated by the orange outline. Right, the CCFv3 atlas. **B** The middle axial section colorized by clusters. **C** Schematic representation of four exemplar paths: including intra-area cross-region (Path1), cross-brain area (Path2 and Path3), and intra-region (Path4). **D** Regional mean features along Path1, Path2, and Path3. We colored the region names with the median feature value of the region. **E** The gradual spatial change in the "variance-percentage" and straightness along the radial direction of Path4 within CP region (left and middle), and voxel values along Path4 on the average brain (right). Confidence interval, 95%. Source data are provided as a Source Data file.

average brain. We quantified this in the profiled features along the four exemplar paths (Fig. 3C–E). Indeed, the intensity profile in the average map along the first three paths does not correlate with the straightness and Hausdorff-dimension features. However, it resembles the "variance-percentage" feature of the microenvironment (Fig. 3D), with

Pearson correlation coefficients of 0.86, 0.67, and 0.71 for Paths 1, 2, and 3, respectively. The top-3 microenvironment features complement each other in characterizing brain anatomy, with small average Pearson correlation coefficients of 0.013, 0.207, and −0.015 for the three paths. Our microenvironment representation appears to be

discriminative for brain parcellation. Similarly, the intensity profile along the radial path of the CP region in the average brain (Path 4) exhibited a linearly increasing trend, aligning with the "variance-percentage" feature of the microenvironment. However, it differs from the decreasing trend observed in the feature straightness of microenvironment (Fig. 3E).

The whole-brain dendritic microenvironments could facilitate the exploration of both inter-regional and intra-regional organization across various brain areas, in addition to the four exemplar paths. Interesting examples include but are not limited to the stereotypy discovered in analyzing the middle sagittal and coronal sections (Supplementary Fig. S4), and the left-right symmetry of feature patterns in two hemispheres of the brain (Supplementary Fig. S4, Figure S5). Overall, the microenvironment analysis is consistent with established brain parcellation in CCFv3, offering finer detail with respect to the dendritic characteristic within each brain region.

### Detecting primary distributions and key morphological variables of fully reconstructed neurons

We next analyzed the fully reconstructed neurons with meticulously annotated axons and dendrites in SEU-A1876. While the neuron reconstructions were manually edited by multiple annotators to ensure the correctness of branching patterns, the limited precision of spatial (3-D) pinpointing in manual annotation caused the skeleton of almost every neuron to deviate slightly from the center of the image voxels of the neurites. To address this, we developed an automatic approach[45] that precisely centered neuron skeletons, facilitating the subsequent analyses of axonal varicosities.

The entire set of SEU-A1876 neurons exhibits a brain-wide distribution, projecting across most major brain regions, with cell bodies in 92 brain regions, primarily located in cortex, thalamus, and striatum (Supplementary Data 6). These neurons span dozens of millimeters (Fig. 4). It has been often observed that different neuron classes are poorly discriminated by global morphology features such as length and branching number[16,46]. To overcome this limitation, we registered the dataset to the CCFv3 using mBrainAligner. The standardization of these neurons' coordinates allowed us to use the spatial adjacency of neurons to augment morphological features, inspired by previous studies[30] and the microenvironment representation (Figs. 1C and 3).

Specifically, we generated a similarity matrix of 47 morphological features of the 1876 neurons, and used the spatial adjacency of neurons as a coefficient matrix to finetune the morphology similarity (Methods, Supplementary Fig. S6). This approach reduced the likelihood of clustering together as the result of potentially incorrect matching of morphological features. Indeed, we were able to produce 4 clusters of full neuron morphologies (Fig. 4A), even if the locations of their somas did not appear visibly separated in 3-D space (Fig. 4C). Visual inspection of examples of neurons in distinct clusters confirmed their difference in appearance (Fig. 4B). Each cluster exhibited intra-cluster diversity, prompting a detailed analysis of subcellular structures as discussed in the subsequent sections. The soma-distribution of the neurons in each cluster indicates that C1 consists of cortical neurons; C2 and C4 contain mostly thalamic neurons and a few cortical neurons; and, most C3 neurons are located in the striatum (Fig. 4C). However, we also noticed that 6%, 25%, 31%, and 11% of neurons innervate from non-dominant brain areas for clusters C1, C2, C3, and C4, respectively. Interestingly, when comparing each pair of the four clusters, the two clusters being compared appeared to be separable even with only three morphological features selected using the mRMR algorithm, although these characterizing features were different in each case (Fig. 4C – lower triangle).

The overall consistency between our de novo clustering outcomes and established primary cell types in the mammalian brain prompted a detailed exploration of the most discriminating features of each cluster (Fig. 4D). We found the most discriminating features vary among clusters (Fig. 4C). At the whole-brain scale, the most prominent features were the "bifurcation distance to the soma" ("bif_EucDist2-soma"), and "remote tilt angles" ("tilt_remote_std" and "tilt_remote_mean", Methods). Importantly, no single feature could separate these four clusters (Fig. 4E), emphasizing that a combination of the top features (Fig. 4D) is necessary to characterize neuron clusters. On average, C1 neurons have a smaller likelihood to have large distal arbors, but typically project over long distances (Fig. 4E, F). Bifurcations of C2 neurons tend to be in close proximity to somas, and a smaller variance of "remote tilt angles" (Fig. 4E). In contrast, C3 neurons rarely have distal arbors, and have a larger variance of "remote tilt angles" (Fig. 4E, F). While C4 neurons correlate with C2 spatially, with comparable branching patterns, they have a substantially greater bifurcation-to-soma distance (Fig. 4C, E and F). Note that C2 and C4 primarily consist of thalamic neurons, the substantial difference between them indicates the potential existence of two major neuron subtypes in these thalamic regions.

### Conserved neuron arborization encodes cortical anatomy

Based on the evidence that fully reconstructed neuronal morphology aligns with neuron class (Fig. 4), we further investigated neurons innervating multiple brain regions. This exploration was twofold: based on (a) the arborization patterns for both dendrites and axons (Fig. 5), and (b) the fiber-projecting patterns that connect these arbors (Fig. 6).

We define sub-neuronal arbors as dense branching sub-trees of full neuron morphologies. Practically the diameter of an arbor can range from about 100 micrometers to millimeters (Fig. 1D). The tight packing in space may indicate putative structural units. Profiling the level of arbor stereotypy provide additional insights to those inferred from full morphologies. We decomposed a single neuron into a series of arbors to obtain the sub-neuronal representation. Manually annotated dendrites were treated as independent arbors due to their obvious layout. We used the AutoArbor algorithm[16] (Methods) to divide axons into multiple internally connected arbors. To facilitate comparison, axons of neurons within the same brain region were decomposed to have the same number of arbors, determined using the majority-vote method for all neurons in the region. Two kinds of arbors, proximal and distal, were defined based on distance from the soma using a threshold of 750 μm (Fig. 5A). The arbors were sequentially ordered by their Euclidean distances to soma (e.g., A1, A2, A3). This method detected 3,776 axonal arbors and 1,876 dendritic arbors. We considered various morphological features (Methods) tailored for the arbor structures, including arbor type (proximal or distal), the volume of the rotated 3-D bounding box of the arbor (μm³), the number of branches, and the Euclidean distance to the soma (dist2soma).

We analyzed arbor features in three brain areas: thalamus, cortex, and striatum. Quantitative assessments across 20 CCFv3 regions highlighted morphological diversity and stereotypy, particularly in axonal arbors (Fig. 5A). Overall, neurons in the cortex and striatum have around 50% proximal arbors, while thalamic regions have an apparently smaller number of proximal arbors. The extent of proximal arbors is also considerably variable in the thalamus, i.e., ventral posterolateral and posteromedial nucleus of the thalamus (VPL and VPM) neurons have more proximal arbors than other thalamic regions. The branching number and the respective maximum density features differ from arborization patterns revealed mostly by the arbor-volume feature, which indicates that several neurons originating in multiple cortical regions have very large arbors. Overall, cortical neurons show larger axonal arbors, and AId and MOs neurons have a clearly larger axonal arbor A2 than neurons in other regions. MOp have smaller axonal arbors A2 than MOs. By contrast, supplemental somatosensory area (SSs) and primary visual area (VISp) neurons have one large axonal arbor A1, which also has a chance to position beyond or below the 750 μm threshold to be either a distal or a proximal arbor. Remarkably, brain regions in the primary somatosensory area (SSp) display

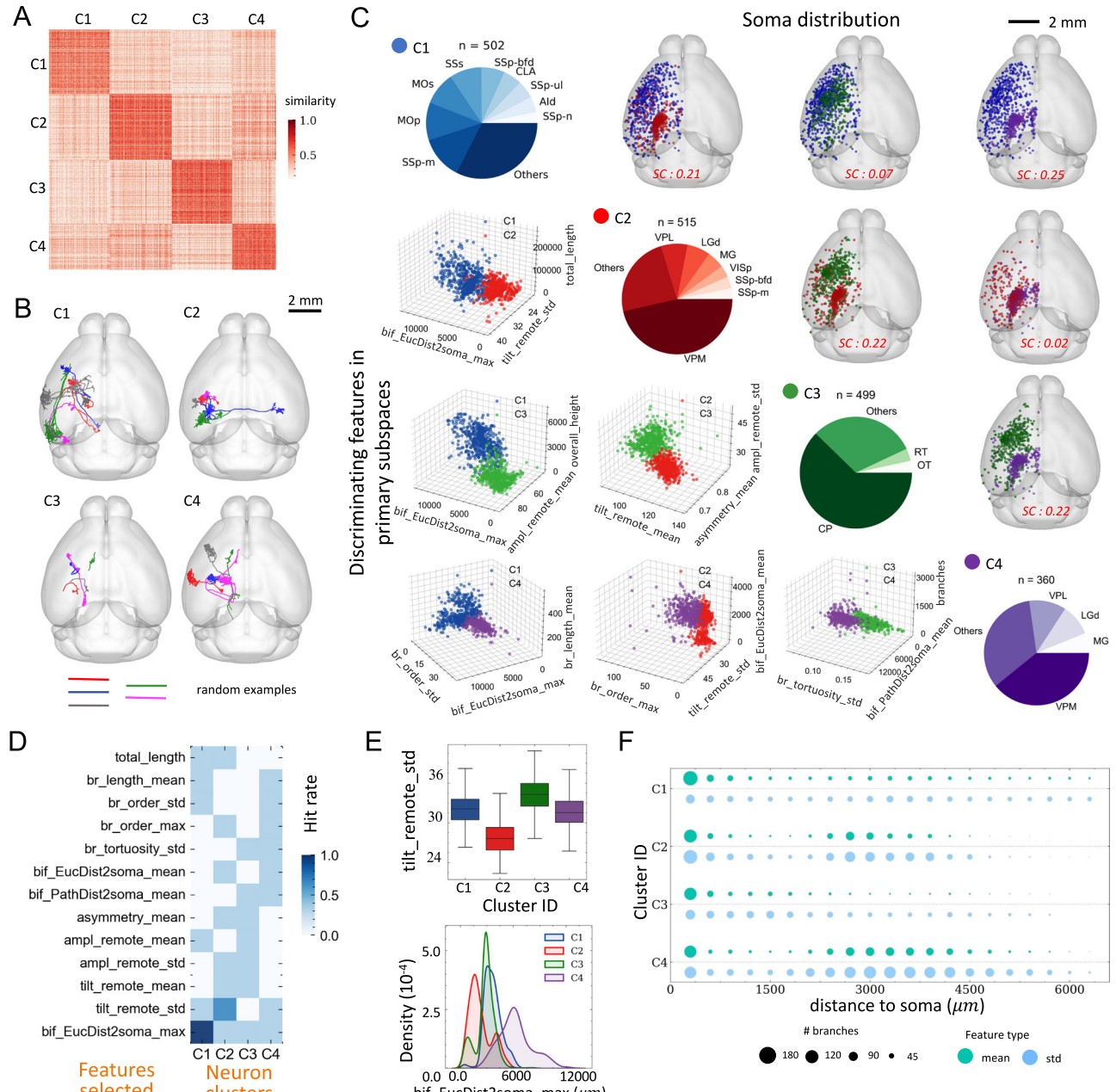

**Fig. 4 | Anatomical characterization of fully reconstructed neurons. A** Heatmap of pairwise neuron similarities. Each row and column are individual neuron, with color showing similarity values calculated as the product of the cosine distance between standardized morphological features over the exponential of normalized between-soma distance. Neurons are categorized into four clusters (C1, C2, C3, and C4) using Spectral clustering (see Methods). **B** Horizontal projections on the CCFv3 template of five randomly selected neurons. **C** A pair-plot displaying the composition of neuron types within each cluster (pie plots in the main diagonal). Soma spatial distributions of cluster pairs are shown in the upper triangle, while 3D scatter plots (lower triangle) show pairwise separability of neurons from each cluster (color-coded) with respect to the top 3 discriminating features between cluster pairs. The average Silhouette Coefficients (SC) are specified in red. Viewpoints of the scatter plots are optimized for cluster separation. **D** Heatmap of the

number of times (hit rate) a feature was selected by mRMR as a top three discriminating feature of the clusters in six independent rounds. Each round corresponded to a separate cluster pair. **E** Top, box plot of the top-ranking feature ("tilt_remote_std") of neurons between clusters. Bottom, density plot of maximal Euclidean bifurcation-to-soma distance across neurons in each cluster. The neuron numbers for C1, C2, C3, and C4 are 502, 515, 499, and 360. **F** Matrix visualization of the mean (light green) and standard deviation (std; light blue) of branch numbers (represented as dot size). Each row corresponds to one cluster, and each column represents the distance interval (300 μm) at which we measured branch numbers. Source data are provided as a Source Data file. Box plot: edges, 25th and 75th percentiles; central line, the median; whiskers, 1.5×the interquartile range of the edges; dots, outliers.

---

dramatically contrasting and indeed combinatorial arborization patterns. SSp-ul and SSp-ll have comparable arbors A1, A2, and A3; however, SSp-m, SSp-n and SSp-bfd have large A2 arbors.

These arborization patterns of cortical neurons, particularly SSp neurons, seem to define a "codebook" that we sought to further examine. We compared arbors within two major cortical projection

classes-extratelencephalic (ET) and intratelencephalic (IT) neurons (Fig. 5B; Supplementary Fig. 17). Differences between projection classes are evident in dendritic features. Indeed, ET neurons have both larger dendrites than IT neurons in the same brain regions. However, IT neurons have higher maximum compartment densities for dendrites. For axonal arbors, ET neurons have smaller A1-arbors, but a greater

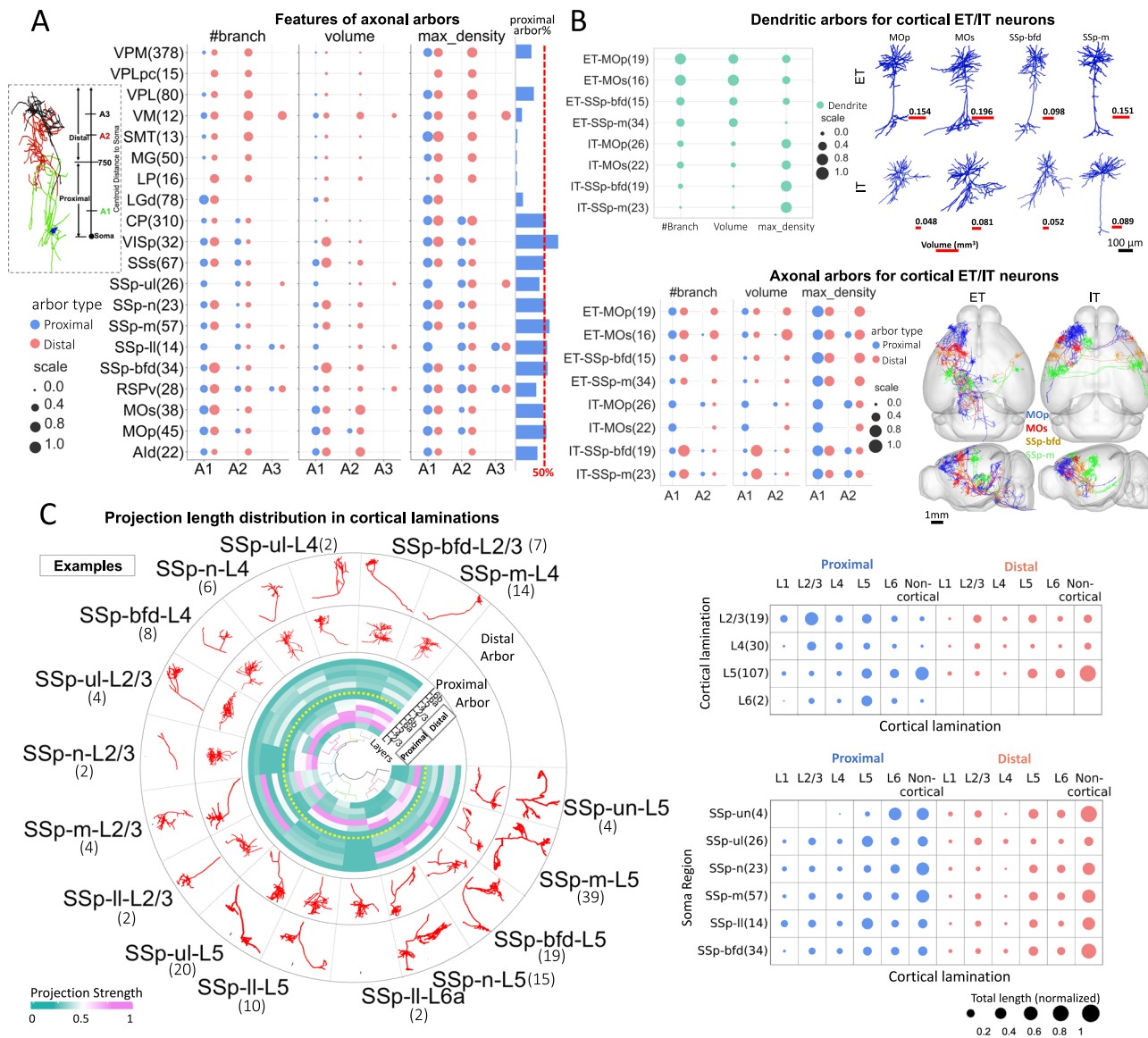

**Fig. 5 | Morphological stereotypy and diversity in neuronal arbors. A** Matrix visualization of normalized morphological features of axonal arbors for 20 soma-types (s-types). The blue and red dots represent the features of proximal and distal axonal arbors respectively, and the ordering of arbors (A1, A2, A3) was determined based on their distance-to-soma values. The top left sketch is an exemplar illustration of the categorization of proximal and distal arbors, and their orderings. The arbor types were determined by their distances from the max density compartment to somas, where a max density compartment refers to the compartment containing the maximal number of compartments within a 20 μm radius. The histogram on the right displays the average percentage of proximal arbors for each s-type. The parenthetical number after region name indicates the number of neurons in that region. **B** Matrix visualizations of normalized morphological features of dendritic arbors (top left) and axonal arbors (bottom left) for extratelencephalic (ET) and intratelencephalic (IT) neurons of 4 cortical regions. The top-right component shows representative dendritic morphologies for each region and projection type. The bottom-right component shows horizontal and sagittal projections of axonal arbors for ET (left) and IT (right) neurons embedded in the CCFv3. **C** Axonal arbor morphologies and projection distributions of lamination subtypes of cortical SSp neurons across cortical laminations. Left, the projection distribution across cortical laminations and their representative structures. The central circular heatmap shows the projection strengths across cortical laminations (radial vectors). The dendrogram in the center of the plot shows hierarchical clustering based on the projection strength heatmap. Two outer layers in the plot show representative examples of proximal and distal axonal arbors. The number of neurons of each subtype is specified in parentheses. Right, matrix visualization of the projection strength for lamination subtypes (top) and s-types (bottom).

chance to have a larger A2 than the respective IT counterparts, consistent with the categorization of these ET-IT neurons.

We also examined the features of neurons in six regions of the primary somatosensory cortex across cortical layers (Fig. 5C). Neurons in the unsigned regions (SSp-un) have large proximal axonal arbors projecting mainly to cortical layer 6 (L6), but not to layer 1 (L1), layer 2/3 (L2/3), and layer 4 (L4), and distal arbors mainly projecting to L5 and L6. Subdividing neurons by laminar position reveals distinct attributes in the projection patterns of proximal and distal arbors, with some overlaps. Axonal arbors of L2/3 neurons primarily project to L2/3 and

L5, while L4 neurons reach mostly L2/3. Instead, L5 neurons project mostly to L5 and L6, and L6 neurons extend projections preferentially to L5 (Fig. 5C). The circular visualization provides a detailed view with soma regions and cortical layer information displayed (Fig. 5C – circular view). It is important to note that this codebook may evolve as more neuron reconstructions become available.

As we observed that thalamic neurons have a variety of arborization patterns (Fig. 5A), we clustered both the morphological features of arbors (8-dimensional) and projection distributions (108-dimensional) of neurons originating from each brain region

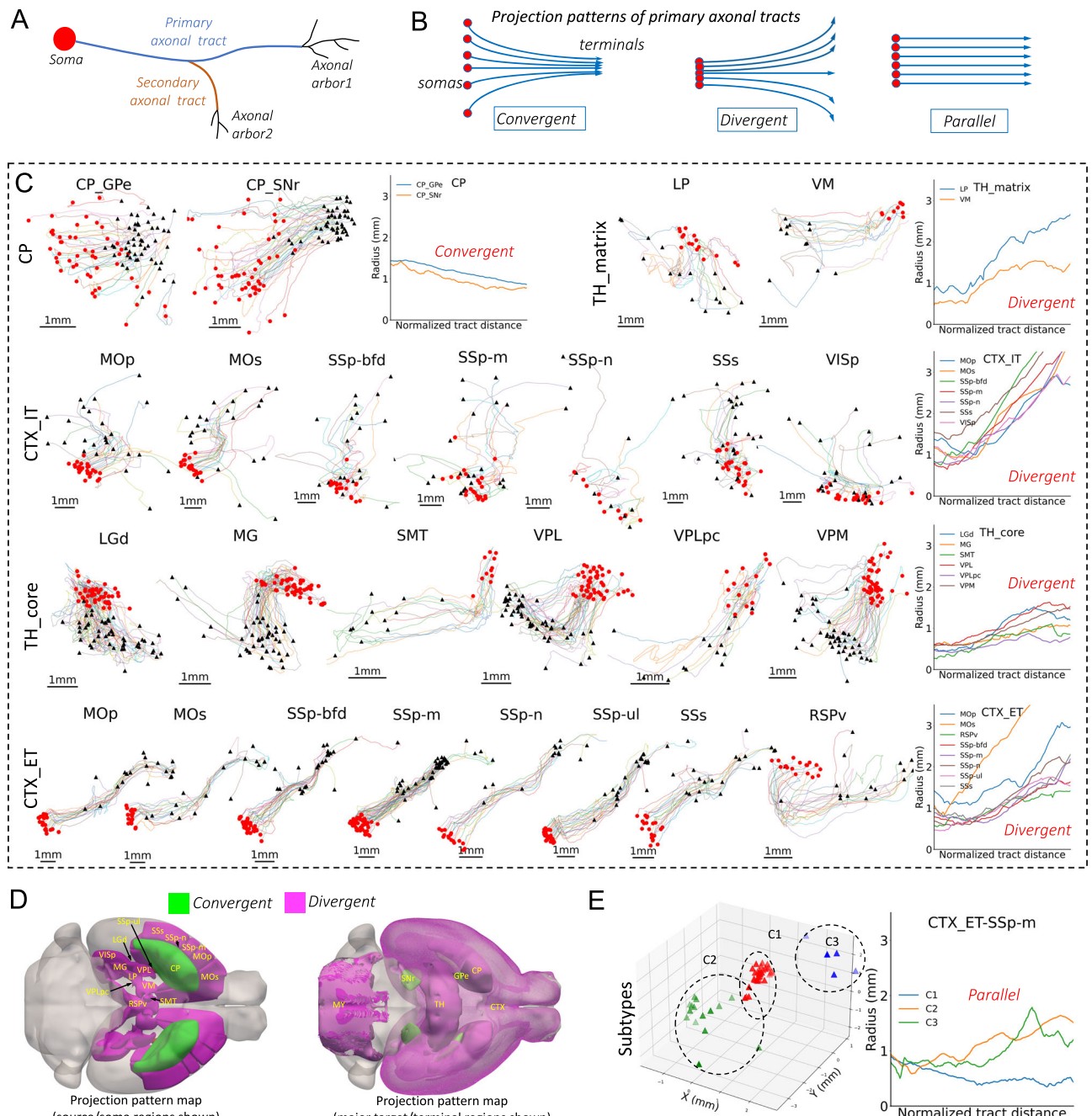

**Fig. 6 | Projection patterns and anatomical insights from primary axonal tracts. A** Schematic illustration showing the axonal morphology, highlighting the blue-colored primary axonal tract, which is the long projecting axonal path after excluding distal short segments. A neuron may contain multiple tracts, such as the secondary tract highlighted in dark orange. **B** Schematic visualization of three distinct projection patterns at the population level: convergent, divergent, and parallel, determined based on the comparative spread in space of somas and terminals. Soma positions are indicated by red dots, while arrowheads denote the terminal points of primary axonal tracts. The blue lines connecting them represent the primary axonal tracts. **C** 2D projections of primary axonal tracts of 25 projection-based subtypes in cortex, striatum, and thalamus. The label on the left specifies the s-type (for CP neurons) or projection classes. Circular red dots represent the somas, while triangular black dots denote the tract termini. In-between tracts are colored randomly. A line plot of the spatial spread (radius) change from the somas to the terminals along the corresponding tracts is appended on the right side for each project type. **D** Horizontal view of projection pattern maps by source (left) and target (right) regions. The regions are colored by the projection pattern type. **E** 3D scatter plot of the terminal point locations for three clusters identified for the L5 ET projecting cortical SSp-m neurons using K-Means clustering based on their terminal points, with the respective spatial spread profiles plotted on the right. The terminal points of the three classes are colored in red (C1), green (C2), and blue (C3). Source data are provided as a Source Data file.

(Methods). Thalamic core and matrix neurons have similar projection volumes overall (Supplementary Fig. S18). In detail, matrix neurons from nucleus of reuniens (RE), lateral dorsal nucleus of thalamus (LD) and ventral medial nucleus of the thalamus (VM) have greater variability in projection volume than neurons from other regions. Morphologically, axonal arbors of thalamic matrix neurons are generally larger and more complex, exhibiting a greater diversity than thalamic core neurons (Supplementary Fig. S18). Indeed, arbors of thalamic core neurons, except LGd, are more conserved in volume. In terms of projections, thalamic core neurons have a higher concentration of arbors in mostly cortical and midbrain areas, which are responsible for sensory and motor control. On the other

hand, thalamic matrix neurons have a wide range of projection targets, covering 108 regions.

## Characterizing motifs of primary axonal tracts

To complement the analysis of neuronal arborization, we further studied the projecting axons connecting major arbors (Fig. 6A). Understanding the diversity and stereotypy of axonal tracts may help to understand the global structure of the brain. We focused on primary axonal tracts, obtained by iteratively pruning short branches off the longest axonal path (Fig. 6A; Methods), and identified three projection patterns, i.e., convergent, divergent, and parallel (Fig. 6B).

In 19 major brain regions with fully reconstructed neurons SEU-A1876, we found different projection patterns (Fig. 6C). First, striatal and thalamic neurons showed opposite projecting tendencies. SNr-projecting CP neurons (CP_SNr) and GPe-projecting CP neurons (CP_GPe) have convergent patterns, with widely distributed somas but tightly packed primary projection targets. The cross-sectional radii tended to decrease from 1.5 mm to sub-millimeters. In contrast, both the thalamic matrix neurons (TH_matrix) and thalamic core neurons (TH_core) show an evident divergent pattern, with somas concentrated in each of the eight thalamic regions, i.e., lateral posterior nucleus of the thalamus (LP), VM, LGd, medial geniculate complex (MG), submedial nucleus of the thalamus (SMT), VPL, parvicellular part of VPL (VPLpc), and VPM, but projection targets wide spread. The cross-sectional radii extended from sub-millimeter to about 1.5 millimeters for TH_core and VM neurons, and reached to the range of 2~3 millimeters for LP neurons.

Different from the striatum and thalamus, cortical neurons showed more complex patterns (Fig. 6C). IT-projecting cortical neurons (CTX_IT) display divergent projections, expanding the cross-sectional radii by about 3 times or more along the primary axonal tracts. However, ET-projecting cortical neurons (CTX_ET) have have a much more conserved axonal trajectories to target brain regions, with deviations only occurring near target regions. Interestingly, the majority of cortical neurons, irrespective of ET or IT projection types, showed a converging pattern at the initial part of the projection pathway, as illustrated by decreased radii immediately after the somas (Fig. 6C).

We also analyzed the topographical organizations for the ET-projecting and IT-projecting cortical neurons, GPe-projecting CP neurons, and VPM neurons, based on the primary axonal tracts (Supplementary Fig. 19). Notably, the termini of primary axonal tracts of ET neurons exhibit a high degree of dispersion within each subtype (Supplementary Fig. S19). However, these termini are conserved across all ET projecting neurons, despite their diverse soma locations (Supplementary Fig. S19). In contrast, IT projecting neurons display distinct topographical organizations, with the termini locations being more closely correlated to their respective soma locations (Supplementary Fig. S19).

We mapped these conserved projection motifs onto CCFv3, with both soma regions and the project target regions highlighted (Fig. 6D). Based on our current data, the brain-wide axonal projects are heavily divergent, regardless of the locations of somas, except for specific cases like CP-SNr and CP-GPe. However, it is also remarkable to see that the divergent CTX_ET projections can be further factorized in terms of clustered target brain regions (Fig. 6C – CTX_ET row). For instance, CTX_ET SSp-m neurons have divergent projections, but their targets can be grouped into three clusters (Fig. 6E, Supplementary Fig. S7). The projection of neurons from each of the three clusters showed a nearly parallel pattern. In other words, the cortical neurons may have a strongly stereotyped, target-dependent projection pattern although overall the diversity is visibly dominant. In this way, these stereotyped projection motifs provide a high-level description of neuronal arbors across the entire brain.

## Cross-scale topography of axonal varicosities

After estimating axonal and dendritic arborizations, we sought to identify putative synaptic sites. As we had analyzed and modeled

dendritic spines in a previous study[47], we used IMG204 to study putative axonal varicosities. Axonal varicosities may be classified as *terminaux* (TEB) and *en passant* (EPB)[48] (Fig. 7A). Using the complete axons in SEU-A1876 neurons, we identified both types of varicosities. To maximize accuracy, we refined manually annotated neuron skeletons with an automated skeleton de-skewing algorithm[45], followed by approximating varicosities using a Gaussian distribution model (Methods; Supplementary Fig. S8). We identified 2.63 million axonal varicosities from all axonal reconstructions (SEU-A1876), averaging 1,404 varicosities per neuron. The identification exhibited high robustness for independently traced but morphologically similar neurons (Supplementary Fig. S21). The high correlation between detected varicosities from such independent sources would not be possible if these varicosities were merely noise without any biological consistency (Supplementary Fig. S21). Benchmark on 1450 manually annotated varicosities showed a high accuracy (99% precision and 91.7% recall). Additionally, EPB ratios of four manually annotated hippocampal CA1 neurons in our dataset are 98.9%, 97.1%, 96.4%, and 97.6%, aligning with electron microscopy-based detection[49]. We also categorized axonal branches into varicosity-branches or null-branches, based on the presence or absence of detected varicosities (Fig. 7A).

We studied the spatial distributions of varicosities at several scales. At the whole-neuron level, we calculated varicosity densities against their distances to somas in 16 brain regions (Fig. 7B). Varicosities of thalamic neurons are predominantly located on the distal axons. Claustrum (CLA) and AId neurons have very broad varicosity distributions. Olfactory tubercle (OT) and RT neurons have high varicosity density along intermediate ranges of axon extensions. Neurons in the other brain regions, including 5 cortical regions and the striatal region CP which has large local axons (Supplementary Fig. S22), have enriched varicosities in local axons (Fig. 7B).

We also generated a varicosity-feature topography for different neurons (Fig. 7C). In each of three major categories of brain areas (cerebral nuclei (CNU), thalamus, and cortex), varicosity feature distributions are typically stereotyped, exception for reticular nucleus of the thalamus (RT) neurons, which have a different feature map from other thalamic neurons. CNU neurons, particularly CP and OT neurons, showed much higher TEB ratios. However, the average patterns across these three brain areas are diverse, offering more detail than the one-dimensional radial distributions (Fig. 7B) that are summarized as the third varicosity feature F3 (Fig. 7C).

In our data, neurons from cortical and thalamic regions have on average 271.1 and 233.8 varicosity-branches, significantly larger than striatal neurons (151.3; Fig. 7D). Notably, the ratios between varicosity-branches and null-branches remain consistently around a value of approximately 3 across neurons from various brain areas (Fig. 7D). Higher varicosity-branch ratios were found in terminal branches than in bifurcating branches such as 81% of the former containing varicosities versus only 71% of the latter. Interestingly, the average lengths of varicosity-branches and null-branches are indistinguishable (Fig. 7D). On average, varicosity-branches of striatum neurons are slightly more curved than null-branches (Fig. 7D). At the branch level, we categorized bifurcating varicosity-branches into three types depending on the type of children branches (Fig. 7E), with a dominance of consecutive varicosity-containing branches (B0 and B1 types, Fig. 7E). These observations suggest that varicosities may aggregate in close-packing axonal arbors. We also found clear differences in the number of varicosities at the individual branch level for various neuron types (Fig. 7D). Furthermore, varicosities are preferentially located at the branch terminal ends (Fig. 7F). Overall, our data suggest that varicosity distribution strongly depends on the scale of analysis: varying dramatically at the full neuron level (global diversity), but sharing analogous patterns at lower structural levels (local stereotypy).

## Characterizing whole-brain diversity and stereotypy using cross-scale features

In observing substantial diversity across different morphometry scales, we questioned whether such diversified patterns across scales could be combined to characterize neurons. To do so, for each neuron, $n_i$, we first concatenated its features across resolution scales (micro-environment, full morphology, arbors, motifs, and varicosities) into a feature vector $f_i$. For two neurons $n_i$ and $n_j$, we obtained Pearson

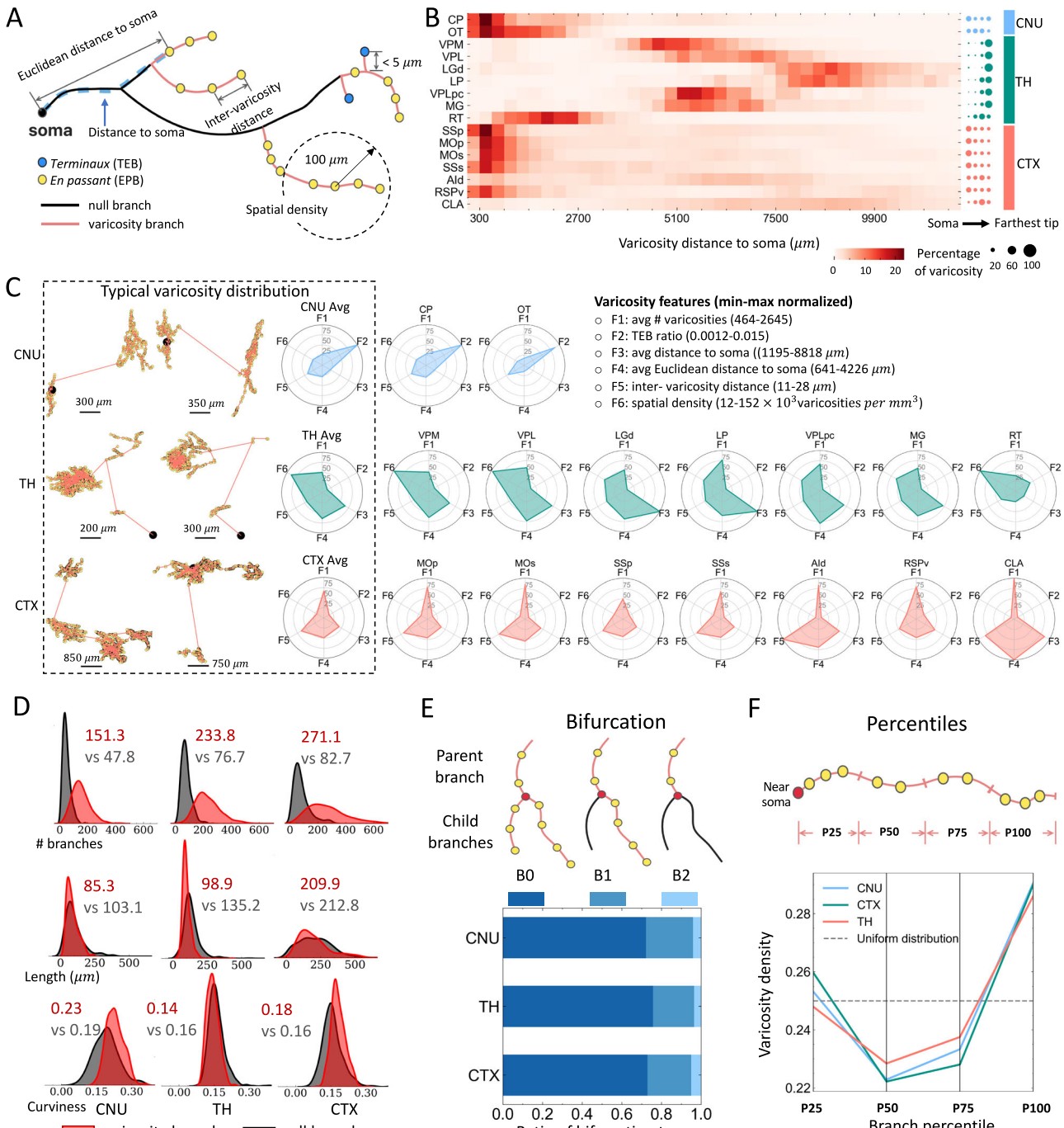

**Fig. 7 | Spatial patterns of axonal varicosities distribution at various scales. A** Varicosity types and key features. **B** Heatmap showing the percentage of varicosities as a function of the distance to the soma. The right panel shows the quartile distribution of varicosities. The distances for each type are normalized independently by the maximal distance among all varicosities for the corresponding type. Labels specify the corresponding brain areas. **C** Within the dashed line frame: the left side shows representative neurons with somas (black) and varicosities (yellow) connected by a minimum spanning tree (MST). The right-side radar charts illustrating the average of six varicosity features, calculated as mean values after min-max normalized to a 0-100 scale. Right, analogous radar charts for each of the s-types within the analyzed brain areas. **D**–**F** Spatial preference of varicosities at various sub-neuronal scales. **D** Density plots of three morphological features between varicosity branches (red, branches containing varicosities) and null branches (gray, branches without varicosities). The feature "length" refers to the path length of a branch, while "curviness" represents the curviness of the branch. The colored numbers are the mean values of the corresponding categories. **E** Top, schematic drawing of three bifurcation types defined according to the presence of varicosities in the two child branches. The parent and child branches are topologically connected, with the parent branches being closer to the soma. Bottom, bar plot of the proportions of the three types of bifurcations in each analyzed brain area. **F** Top, schematic drawing of the length quartiles of a varicosity branch. Bottom, line plot of the ratio of varicosities distributed at quartiles of a varicosity branch. The horizontal dashed line represents the expected distribution if varicosities were uniformly spaced. Source data are provided as a Source Data file.

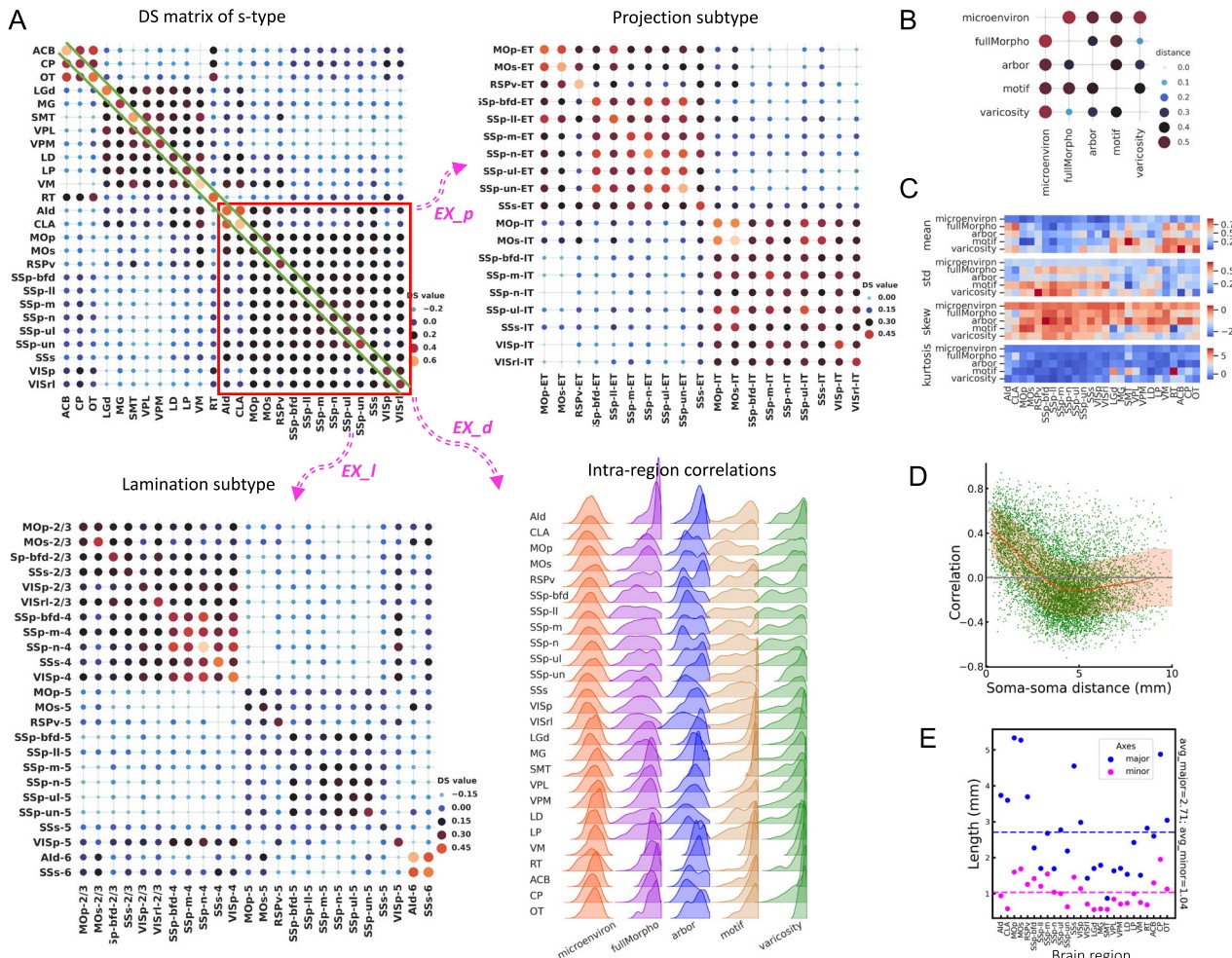

**Fig. 8 | Quantitative diversity and stereotypy analyses based on cross-scale features. A** The Diversity-and-Stereotypy matrices (DS matrices) for s-types (upper left), projection subtypes of cortical neurons (upper right), and lamination-based subtypes of cortical neurons (bottom left). Each value in the matrix (DS value) is the average correlation between all neuron pairs of the two corresponding cell types. The diagonal values are the intra-region average correlations, and the others are inter-region average values, representing intra-region stereotypy and inter-region diversity respectively. The correlation is the Pearson correlation coefficient between cross-scale features, which are the concatenation of standardized features from 5 morphological scales: microenvironment ("microenviron"), full morphology ("fullMorpho"), arbor, primary axon tracts ("motif"), and varicosity. Bottom right, density plots of the distributions of intra-region correlations for various s-types at different morphometry scales, that is, the distribution of diagonal items in the left component of (**A**). **B** Pairwise distances between the DS matrices of different scales. The distance is obtained by computing 1 minus the Pearson correlation coefficient of the DS matrices. **C** Heatmaps of the first, second, third, and fourth orders of statistics of the intra-region correlation distributions for each morphological scale (bottom right of (**A**)). **D** Scatter plots showing the relationship between soma-soma distances and the correlations of the cross-scale morphometry features. Linear correlations are observed when the pairwise distances are small. The red lines and red shadows within the boxes represent the means and correlation ranges within one standard deviation (σ) around the mean values. **E** Scatter plot of the major and minor axis lengths of brain regions. The dashed lines are the average lengths for the major and minor axes of all regions. Source data are provided as a Source Data file.

correlation of the concatenated features of two neurons, $c_{ij}$, to measure the similarity. Next, for neurons innervated from two brain regions $U$ and $V$, i.e., two soma-types (s-types), we averaged the correlation coefficients of all inter-region neuron pairs to produce an overall cross-scale feature similarity score $s_{UV}$ in these two regions. A $s_{UV}$ score close to 0 indicates minimal commonality between neurons in the two regions, while scores approaching 1 or -1 indicates high similarity or dissimilarity, respectively. When U and V are the same region, the score becomes $s_{UU}$ (or $s_U$ for simplicity), which measures the intra-region averaged similarity, or "intra-type" stereotypy. In this way, we constructed a Diversity-and-Stereotypy (DS) matrix $S$, where each entry is $s_{UV}$, for all pairs of brain regions to quantify the distribution of morphological patterns (Fig. 8A).

We found that cross-scale features were able to discriminate between different neuron types. Indeed, the DS matrix of all soma-types showed three clear modules, which correspond to the majority of cortical, thalamic, and striatal neurons (Fig. 8A – top-left), except

for the thalamus reticular nucleus (RT) neurons, which are distinguishable from other thalamic neurons in terms of neurotransmitters and connectivity[50]. In addition, the DS submatrix of cortical neurons correlates negatively with that of the thalamic neurons, but exhibits weak correlation with the striatal neurons. Thalamic neurons also correlate weakly but also negatively with striatal neurons. Within each module, DS values are relatively large with small variations, indicating that neurons are remarkably conserved in these brain regions. This modular grouping of brain regions based on cross-scale features is also consistent with our alternative analyses, e.g., microenvironment analysis (Fig. 3).

We focused on the diagonal of the DS matrix (Fig. 8A – EX_d) to examine the distribution of features for the five resolution scales (Fig. 8A, – intra-region correlations, Supplementary Fig. S9). Although overall cortical, thalamic, and striatal neurons have similar average DS scores within each brain region (mean-values = 0.34, 0.47, and 0.52, respectively, as shown as the diagonal values in Fig. 8A – DS matrix of

s-type), they have different degrees of stereotypy with respect to morphological scales. For instance, for microenvironment features, the average correlation value of thalamic neurons (0.29) is much larger than that of cortical neurons (0.07) (Fig. 8A – Intra-region correlations), indicating that microenvironment features are more discriminating for thalamic neurons. Similarly, certain cortical neurons, like CLA neurons, display high conservation in full morphology and varicosity features, indicated by high mean correlation values (0.8 and 0.7, respectively) (Fig. 8A - Intra-region correlations).

We also used the DS matrix to examine subtypes of neurons. We focused on two subtypes, i.e., neuron-projection subtypes (Fig. 8A - EX_p) and soma-lamination subtypes (Fig. 8A - EX_l) for cortical regions that contain at least 10 fully reconstructed neurons. Specifically, our analyses included 1513 neurons for s-type, 1350 neurons for projection subtypes, and 431 neurons for lamination-based subtypes of cortical neurons. For projection subtypes (Fig. 8A - EX_p), most DS scores among ET neurons are larger than 0.3, which also holds true for IT neurons. However, the majority of ET neurons correlate weakly with IT neurons, even when they are located in the same brain regions (e.g., SSp-n-ET vs SSp-n-IT neurons). Interestingly, several projection subtypes such as MOp-IT, MOs-IT, SSs-IT, and SSs-ET neurons show considerable correlations with all neuron subtypes. The DS matrix also highlights an interesting submodule composed of six SSp ET projecting subtypes, with pairwise correlations higher than 0.4 in most cases.

We observed modularity for cortical laminar subtypes (Fig. 8A - EX_l). L2/3 and L4 neurons are inter-correlated with each other, but exhibit weak correlation with other layers, with clear module boundaries. In the module of L2/3-L4 neurons, a sub-module consisting of five L4 subtypes, SSp-bfd-4, SSp-m-4, SSp-n-4, SSs-4, and VISp-4, also stands out, with a DS score around 0.4. L5 subtypes also appear stereotyped in the DS matrix, but inter-region correlations tend to be weak, in the 0.15 range. The two L6 subtypes, AId-6 and SSs-6, highly resemble each other but have slightly different correlation profiles with other subtypes. Interestingly, VISp-5 neurons show negative correlations with most of the L5 neurons and all L6 neurons, but correlate considerably with L4 and L2/3 neurons. In addition, neurons from the same brain region but in different layers are not necessarily correlated. For instance, the L5 subtypes of SSp neurons and the respective L4 subtypes are negatively and weakly correlated.

We also attempted to understand the relationship among features of different scales. To do so, we calculated the "distance" between each pair of scales (Fig. 8B, Methods), along with the statistics of these features for different brain regions (Fig. 8C). We found that microenvironment and motif features were far away from features of other scales. Instead, varicosity features had small distances to both full morphology and arbor features (Fig. 8B). Therefore, microenvironment and motif features have relatively little redundancy when combined with other scales to categorize neurons and brain regions. The two separate pairs of scales, i.e. {full morphology and varicosity} and {arbor and varicosity}, could be used to cross-validate whether or not data analyses are consistent across scales.

Our analysis above, especially the DS matrices of the projection and lamination subtypes of cortical neurons, indicate that neuronal types can be well defined by their axonal projections and soma location (Fig. 8A). This suggests an underlying relationship between spatial distribution and morphogenesis, indicating proximal neurons sharing more similar morphologies. We tested this hypothesis by evaluating the correlation between morphological correlation of neurons and soma-to-soma distance. The morphological similarity between neurons was negatively correlated with both the soma-to-soma distance, within a scale of 4 millimeters, comparable to the sizes of brain regions (Fig. 8D, E). This correlation aligns with the morphological similarity observed among neurons within the same region (Figs. 3, 5–7, and 8A). Simultaneously, it reinforces the inclusion of spatial adjacency in

microenvironment construction (Fig. 3) and single neuron clustering (Fig. 4).

Another approach to integrating cellular morphometry across scales involves iterative modularization of neuromorphometry. To illustrate, we examined the relationship between projection patterns (represented by "Delta Radius" (Supplementary Fig. S23), the difference in radius between the termini of primary axonal tracts and their somas) and dendritic arbor features of different cortical neurons. Both the "volume" and "max_density" features demonstrated a linear correlation with the radius difference for both ET and IT neurons, with the absolute values of Pearson correlation (R) greater than 0.5 (Supplementary Fig. S23). Specifically, the dendritic arbor volumes in ET neurons showed a strong positive correlation with "Delta Radius" (R = 0.94, P = 0.0019), suggesting that ET neurons with larger dendrites tend to have more divergent projections. In contrast, dendritic arbor volumes in IT-projecting neurons were negatively correlated with "Delta Radius", indicating that larger dendritic arbors are associated with more convergent projections. Additionally, both ET and IT neurons displayed positive correlations (R = 0.66 and 0.51, respectively) between the "max_density" feature and the radius difference (Supplementary Fig. S23). To underscore cellular diversity across the detected modules, we also provided visual examples (Supplementary Fig. S24). Notably, neurons in modules M10 and M12 lack significant local axonal arbors, unlike in modules M5, M6, and M7, where local axonal arbors are present. Moreover, neurons in module M5 exhibit larger volumes compared to those in other modules (Supplementary Fig. S24).

## Discussion

We studied the morphological patterns of neurons in the context of whole mouse brains at multi-scales, from centimeters to sub-microns, with specific focus on the quantification of the diversity and stereotypy of neuronal structures (Supplementary Fig. S25). We leveraged the collaborative effort of the BICCN community to collect and standardize one of the largest mammalian brain imaging databases to the latest Allen Common Coordinate Framework, followed by systematic extraction of morphological features from whole brain level to axonal varicosity level. Subsequently, we categorized morphological patterns in the cortex, striatum, and thalamus, in conjunction with their soma-distribution, projection trajectories and targets, and more detailed arborization and detected varicosities when applicable. Using rich representations of morphological data, we discovered brain modules and morphology motifs across scales, and identified the suitable spatial scales for quantifying the diversity and stereotypy of morphological patterns.

Our multi-scale analysis attempts to complement a number of previous efforts in generating macroscale, mesoscale, and microscale morphometry in the mouse brain and other model systems[51–56]. At the neuron-population level, we analyzed the modular organization of brain regions based on neurite distribution patterns. Previously, modules of mammalian brains have been studied in macroscale, primarily using functional Magnetic Resonance Imaging[57], and in mesoscale, such as the brain-wide neuronal population based projecting-networks using whole-brain optical imaging[53,54,58]. Our analysis confirmed several previous observations such as neighboring regions being more likely in the same module[54,57]. We also additionally estimated modularization from large-scale analysis at the micron and even sub-micron resolutions. This study represents a notable advancement beyond our previous work on single-scale, straightforward neuron morphology screening for mouse brains[16] and other model systems[4]. These earlier studies did not analyze the vast array of patterns observable at and across different scales. In contrast, this study expands the scope and delves deeper into the complex interplay of neuronal patterns at multiple scales, offering a more complete picture of neuronal morphology.

We constructed dendritic microenvironments to enhance the ability to discriminate the structure of local dendrite arborization. Historically, the morphological features of local dendrites were thought to offer limited power for discriminating neuronal classes[59,60]. These observations have also motivated recent studies that rely on fully reconstructed long axons to differentiate neuron classes[15–17]. Nonetheless, the cost to produce long axons or full neuron morphology is still high, and sometimes is exceedingly expensive for large mammalian brains such as primates[61]. We have recently proposed aggregating the spatial neighboring information of local dendrites of human cortical neurons with their 3-D morphology, and thus have obtained superior classification performance of neurons[30]. In this study, we followed the same principle to formulate dendritic microenvironments that offer a valid alternative to integrate spatial information of neurons and their morphology. The microenvironment representation of a large set of dendrites allows for the visualization of the covarying morphological features of neighboring neurons, thus providing a greater chance to differentiate neurons that have limited dendritic features to discriminate each region. This aligns with the finding that neurons in different cortical regions of the human brain share cell types but in different proportions[62]. Meanwhile, the ensemble nature of it helps alleviate the possible imperfections of reconstructions. It provides a balanced compromise between the scarcity of available single morphologies and the limited discriminatory capacity of soma density. Our approach has allowed visualization of more anatomical detail for several brain regions compared to what had been documented in the CCFv3 atlas[26] and the Mouse Brain in Stereotaxic Coordinates[63].

In addition to introducing dendritic microenvironments, we were able to identify critical, minimally redundant factors that contribute to the different categorizations of individual neurons, for their full morphologies. We found that the clustering of cortical, striatal and thalamic neurons into broadly recognizable clusters, each with a specific fingerprint, could emerge with little a priori knowledge. The key features could be identified in the least redundant subspace of spatially tuned morphology features. This finding also complements the conventional parcellation of brain regions in anatomical atlases primarily based on cell densities. Future studies in this direction, potentially combined with the microenvironment analysis of neurites, might suggest alternative approaches to investigate the murine brain anatomy using morphological, physiological, molecular and connectional properties of neurons[2,23].

Individual neurons have traditionally been studied by analyzing their overall morphology[64,65]. However, it is intriguing to explore the variability of arborization and projection patterns within neurons, as they naturally constitute interconnected sub-trees and projecting neurite tracts. We note that this aspect has not been extensively investigated to date. To address this, we undertook a decomposition of single-neuron morphologies into densely packed sub-trees, referred to as arbors. These arbors serve as structural foundations for potential neuronal functions. Additionally, we categorized the arbors according to their proximity to the respective somas. Furthermore, we extracted the primary projecting tracts of neurons originating from different brain regions and examined their spatial divergence and convergence patterns. This approach simplifies the comparison of different neuron types while retaining crucial morphological information. Moreover, it facilitates the quantification of the diversity of conserved patterns, denoted as "motifs" of arbors and neurite tracts. Our work complements previous endeavors aimed at characterizing sub-neuronal structures, such as branching topologies[66,67].

The investigation of synaptic connectivity is a contemporary and critical topic. While electron microscopy remains the gold standard for synapse identification, its limited range (-1 mm$^3$) currently prevents its applicability to mammalian brain-wide axonal projections. Previous studies have thus focused on detecting and analyzing potential synaptic sites collected by optical microscopy[68–70] using various labeling techniques, including genetic or antibody labeling for pre-synaptic and/or postsynaptic sites, as well as a combination of both[47,71,72]. This study aims to expand on existing synapse-detection research in three ways. First, the full morphologies of nearly 2,000 neurons were used to provide a high-quality dataset for analysis. Second, whole-brains, encompassing a number of cortical, striatal, and thalamic regions, were used to provide a complete picture of the distribution of putative synaptic sites. Third, we explored a wide range of features associated with putative synapses. In this way, we have characterized the patterns of brain-wide varicosity-distributions across various cell types that complement previous studies. Of note, while the biological validation of the predicted axonal varicosities is beyond the scope of this resource study, we have utilized statistics from independent yet morphologically similar neurons in the same brain regions (Supplementary Fig. S21). The distributional consistency demonstrates that it is highly unlikely for the predicted varicosities and their patterns to lack biological relevance.

The knowledge gathered from investigating various spatial scales prompted us to develop an integrated model of neuron morphometry and brain anatomy. As an initial effort, we introduced a DS matrix to measure the degree of diversity across neurons with respect to the stereotypy of neuron types. We observed interesting hierarchical and modularized organization of neurons in cortical, striatal and thalamic regions emerging in a quantifiable way, even without explicit clustering. This finding has two valuable implications. First, it confirms complex neuron morphology strongly correlates with existing brain anatomy in the established mouse brain atlases such as CCFv3. Second, and more importantly, it allows us to hypothesize that for a more complicated mammalian brain such as those of primates, an effective way to explore and understand the brain anatomy and even the associated brain functions could take a similar multi-scale approach, instead of relying solely on anatomists' manual drawing of brain structures. The present study highlights the power of large scale systematically mapped neuronal data in elucidating detailed cell type structure and morphology. Our cross-scale integration of information may also extend to incorporate in the future other data modalities such as single-cell transcriptomic data[59,73,74].

Many of our observations align well with previous studies, including the similarity between the projection patterns calculated from primary axonal tracts (Fig. 6) and those estimated from single neuron morphologies[16]. The regions in most detected modules (Fig. 2; Supplementary Data 5) is consistent with experiments[33,38], and the proportion of EPB varicosities is similar to that observed in electron microscopy studies[49]. On the other hand, there are many previously unexplored findings (Supplementary Data 8). One such example is the discovery of three subtypes for the L5 neurons in the primary somatosensory area - mouth region (Fig. 6E, Supplementary Fig. S7), based on the clustering of primary axonal tract termini. Another finding is the identification of four primary clusters for all single neuron morphologies. We introduced spatial adjacency into feature comparison, allowing unambiguous identification of four large, primary clusters with clear separation (Fig. 4). Within each cluster, neurons exhibit substantial diversity, quantified in this work to measure stereotypy and separation between clusters. These in-cluster variance motivates the identification of sub-neuronal conserved structures for characterizing neurons at finer scales.

These data-driven findings, resulting from correlation analyses, warrant dedicated experiments to unravel functional mechanisms. As a resource paper providing morphometry data and analytical tools, the verification of these findings is beyond its scope. Nevertheless, these insights offer valuable directions for future biological experiments, making the resources in this work a valuable mining-and-

validation protocol for the neuroscience community. In our ongoing exploration of these resources, we aim to delve deeper into the morphological diversity between different scales, as well as adapt to a broad range of neurons. For example, while our current dataset primarily comprises projection neurons (Supplementary Data 6), we acknowledge the importance of exploring interneurons, which constitute 20%–30% of the neocortex in the human brain[75]. We plan to incorporate public-domain reconstructions of interneurons[76] mapped to the CCF, enabling a joint analysis with our datasets. Additionally, we aim to investigate correlations between conserved patterns at various scales, exploring aspects such as cellular diversity across different modules and the relationship between axonal arbors and projection patterns.

## Methods

### Nomenclature

The nomenclature of brain regions follows the CCFv3[26], which categorizes a mouse brain into 671 regions. Each region, except for the direct tectospinal pathway (tspd), comprises two mirroring subregions in the left and right hemispheres. A higher level of granularity consisting of 314 regions (CCF-R314) is used by merging highly homogeneous regions, such as the lamination-differentiated cortical subregions. All brain regions used in this work are from the CCF-R314 regions. We have spelled out the full names of the regions in the manuscript whenever we refer to them for the first time. The CCFv3 atlas can be found at https://connectivity.brain-map.org/3d-viewer?v=1.

Super-regional anatomical entities, such as brain areas, are sets of functionally related regions that are continuous in space. While the definitions of brain areas are similar, they differ in granularity. In this paper, we discussed a higher granularity consisting of 4 areas: cortex (CTX), cerebellum (CB), cerebral nuclei (CNU), and brain stem (BS). We also discussed 13 compound areas, which are CBN: cerebellar nuclei, CBX: cerebellar cortex, CTXsp: cortical subplate, HPF: hippocampal formation, HY: hypothalamus, isocortex, MB: midbrain, MY: medulla, OLF: olfactory areas, P: pons, PAL: pallidum, STR: striatum, and TH: thalamus.

To facilitate understanding, specific terms describing multi-scale morphometry and morphological patterns are detailed in Supplementary Data 7.

### Image acquisition and processing

We collected 204 whole mouse brains at submicron or micron resolutions from 4 BICCN projects within the BICCN community and another collaboration project. Of these, 180 fMOST brains came from a U19 project (1U19MH114830-01). The other 10 fMOST brains and 10 STPT brains were obtained from another U19 project (1U19MH114821-01) and 1 LSFM brain from a U01 project (1U01MH114829-01). These brains were downloaded from the Brain Image Library (BIL, http://www.brainimagelibrary.org). 3 LSFM mouse brains were provided by P.O. (n = 2) and Z.W. (n = 1), who were granted from another U01 project (1U01MH114824-01). The brain images exhibit anisotropic resolutions, primarily ranging from 0.2 to 0.35 μm in the xy plane and 1 μm in the z direction. The fMOST images were labeled using sparse viral-like transgenic lines. The primary reporters include TIGRE-MORF (Ai166), GFP-expressing TIGRE2.0 (Ai139 or Ai140), and TIGRE1.0 (Ai82). These reporters were paired with various drivers, such as Cre expression lines, which target distinct yet specific neuronal types[16]. Detailed information regarding brain IDs, modalities, sources, resolutions, downloadable links, transgenic labeling and the primary targeted neuronal types is summarized in Supplementary Data 1.

The brain datasets, totaling 3.7 peta-voxels, are managed via MorphoHub[77]. To get fast access to both fine-grained details and a global overview, we restructure each brain into hierarchical TeraFly[78] format.

### Registration

Brain images were registered to the 25 μm CCFv3 template using the cross-modal registration tool mBrainAligner[27,28]. We employed a similar pipeline consisting of image standardization and preprocessing, global registration, and deep learning enhanced local registration, with a minor update on the landmark searching strategy which improved overall registration accuracy for the hippocampal and striatal neurons. Registration channels were leveraged whenever possible. The brains were down-sampled to approximately 25 μm resolution through even-folds linear interpolation prior to registration. Non-brain tissues were semi-automatically removed with Vaa3D[79–81] and mBrainAligner. Anatomical regions of the brains were automatically labeled based on the CCFv3 atlas and the deformation matrices obtained during registration. The multi-morphometry including full morphologies, local morphologies, arbors, and varicosities, were reverse-mapped to the CCFv3 space using the inverse deformation matrix. We estimated the robustness of the registration by computing the average registration offsets and intensity variance across 191 analyzed brains for 2213 landmarks (Supplementary Fig. S10), sampled homogeneously from the regional boundaries of CCFv3 atlas. For each landmark, the intensity variance was normalized by the average offset. The CCF-space average brain was generated by computing the mean voxel value for each voxel on the CCFv3 atlas across all 191 brains.

### Soma identification

The SEU-S182K soma dataset comprises two parts. The first part consists of 45,664 somas manually annotated using the Vaa3D-TeraFly platform, most of which were reported in a previous study[77]. An additional 136,833 somas were semi-automatically annotated using the Collaborative Augmented Reconstruction (CAR) platform[29], which is updated from our Vaa3D-TeraFly and immersive annotation system Vaa3D-TeraVR[82].

The semi-automatic soma identification protocol involves two major steps. Firstly, we filtered out the TeraFly image blocks (-256 voxels in each dimension, -59 × 59 × 256 μm$^3$) with maximal intensities less than 250 (unsigned 16-bit image), standardized the remaining blocks through a Z-score normalization, and converted them to the unsigned 8-bit range. Next, the blocks were binarized using their 99th percentile as thresholds, and the resulting images were transformed using the gray-scale distance transform (GSDT) algorithm. Candidates were identified as voxels with intensities in the range of 5 to 30 on the transformed image, followed by a Non-Maximal-Suppression (NMS)-like strategy to eliminate redundant candidates.

In the second step, we cropped 128 × 128 × 128-sized image blocks on the second-highest resolution images (-59 × 59 × 256 μm$^3$) centered at the position of putative somas. These blocks were then distributed to remote users on the mobile application CAR-mobile. Using this protocol, we identified 179,115 somas within weeks, involving 23 trained annotators and 7 fresh annotators without any prior knowledge.

The soma locations were then optimized by applying the mean-shift algorithm bound with Vaa3D after zero-clipping of voxels with intensity lower than $\mu + \sigma$, where $\mu$ and $\sigma$ are the mean and standard deviation of the image block. A window size of 15 voxels was used for the mean-shift soma location optimization. Possible duplicates were removed when two somas are within 15 voxels and their center point intensity was lower than the average intensity of the two somas. Somas outside of CCF-R314 regions were excluded.

### Neurite detection

Neurite voxels were segmented using an automatic algorithm and then summarized the number of voxels by brain regions to produce a region-wide vector consisting of 314 values. Each value represented the total number of detected neurite voxels in the corresponding brain region, normalized by the total number of neurite voxels in the brain. The algorithm segments neurite in 191 out of the 204 brains on low

resolution images, averaging at approximately 1 μm ($x$) × 1 μm ($y$) × 4 μm ($z$), for a trade-off between accuracy and efficiency. Among these, 183 brains were from 34 genetically labeled driver genes (Supplementary Fig. S3). Consequently, each of the 314 regions was represented by a brain-wide neurite distribution vector, resulting in a neurite density matrix ($M_d$) with dimensions of 314×191. The neurite density for each region was computed as the total number of detected neurite voxels within that region divided by the total number of neurite voxels detected in the entire brain. Using the matrix, we estimated a pair-wise regional correlation map ($M_c$) with dimensions of 314×314, among which each value was the Spearman correlation coefficient between the normalized 191-dimensional vectors of the corresponding region. To counteract the imbalance in the number of brains across different transgenic lines, we employed normalization during the estimation of correlation coefficients using the "wCor" package in R, where each brain was assigned a weight that corresponds to the reciprocal of the number of brains from the respective transgenic line. In cases where brains without transgenic labeling information, we used the average of all weights.

Based on the correlation map ($M_c$), we assessed the intra-compound area consistency as the distribution of correlations between all pairs of regions within the compound area. In this way, the distributions for all 13 compound areas were calculated (Fig. 2A).

The neurite detection algorithm is an efficient pipeline without utilizing deep learning models. The procedure can be summarized in 6 steps:

1. Estimation of an empirical foreground threshold for each brain. This step involved finding an empirical threshold value (*thresh*) to distinguish between the foreground (neurite voxel) and background (non-neurite voxels) on the lowest-resolution image. The threshold was estimated based on the mean and standard deviation of all voxels: $0.9 \times \min(\max(\mu + 1.5\sigma, 400), 1000)$, where $\mu$ and $\sigma$ were the mean and standard deviation of all voxels of the brain (16-bit image).

2. Split the brain into non-overlapping image blocks. In this step, the third lowest resolution brain images were split into small non-overlapping blocks for subsequent memory-affordable processing. We utilized the image blocks of Vaa3D-TeraFly files at the specific resolution of approximately 256 voxels in each dimension.

3. Pre-filtering. This step involves filtering out blocks that are unlikely to contain neurite. Image block files (in compressed TIFF format) that were smaller than a certain size (1.7 MB) or had a maximum pixel value lower than 300 were considered neurite-free blocks and were excluded.

4. Calculation of the salient map. An image block was firstly denoised by an adaptive filter similar to the "ada_threshold" plugin on Vaa3D. At the same time, an anisotropic salience map was estimated through block-wise PCA analysis on 16 × 16 × 16 voxels-sized cuboids which were upsampled from 16 × 16 × 4 cuboids of the original image, as the resolution in $z$ axis of the image is about 3 times smaller than that in $x$ and $y$ axes. The anisotropy score of each cuboid is defined as $\frac{S_1 - S_2}{S_1 + S_2} \cdot \frac{S_1 - S_3}{S_1 + S_3}$, where $S_1$, $S_2$, and $S_3$ are the eigenvalues of first, second, and third principal components of the cuboid, similar to the content index Q in previous studies[83]. The final salient map was calculated by multiplying the denoised image and anisotropy map.

5. Thresholding. The salient map was thresholded using $0.1 \times thresh$ calculated in step 1. *thresh* may be manually adjusted based on the segmentation results if it was inappropriate.

6. Mapping neurite voxels to CCFv3 atlas. Finally, the identified neurite voxels were mapped to the CCFv3 space and summarized by regions to obtain a region-wide neurite voxel vector in the shape of 314. The vector was then normalized by dividing it by the total number of neurite voxels, resulting in a neurite density vector for the brain.

Our neurite detection algorithm was qualified by the good linearity between the number of annotated somas and the total number of detected neurite voxels of each brain and in each region (Supplementary Fig. S3C). The considerable variety of sparse labeling and a large number of transgenes ($n = 34$) and brains ($n = 191$) provide a minimal-redundant neurite matrix, laying the foundation of a reliable neurite detection. A large set of diverse labeling genes and brains down-weights the co-expression patterns and highlights possible morphological relationships. When two regions show a high correlation between their distributions across 191 brains, it may indicate a possible connection. To estimate the possibility of mislabeling neighboring regions, we calculated correlation coefficients between the number of annotated somas in neighboring regions for the 10 brains with the highest annotated somas. The coefficients were small, ranging from 0.001 to 0.2797 (Supplementary Fig. S12). Neurite detections on image blocks were presented in Supplementary Fig. S3A and Supplementary Fig. S11A. An example of the neurite connections and corresponding detections between the regions CP and GPi/GPe was illustrated in Supplementary Fig. S11B.

### Target-correlated regions detection

We considered each region as the target region and extracted regions having a Spearman correlation coefficient of no less than 0.5 with it, thereby forming its highly correlated region set. Out of the 314 regions, 313 regions have highly correlated regions based on this criterion. A detailed list of these region sets is provided in Supplementary Data 4. We classified the region sets into two categories: intra-compound area (intra-CA) and cross-compound area (cross-CA), based on whether the regions were within the same compound area.

### Regional module detection

We classified all regions into 18 non-overlapping subsets or initial modules, based on the dendrogram produced by applying hierarchical clustering to the correlation map ($M_c$). The module detection process begins by finding a seed branching point, which is the cluster with the lowest level (*i.e.*, the first diverging cluster) in the dendrogram, followed by checking the number of all its subsidiary region leaves. If a cluster contains no less than 3 regions and no more than 30 regions, it is defined as an initial module. Otherwise, we merge the current cluster with the most closing cluster or split it into modules. The process repeats until all regions are categorized into a module, resulting in a total of 18 non-overlapping initial modules.

For all the initial modules, we removed any region that occurred less than twice in the target-correlated region sets (Supplementary Fig. S13; Supplementary Data 4). If a module had at least two regions remaining, it was considered a tight module. Otherwise, it was deleted. Using these criteria, we identified 18 tight modules (Fig. 2B, Supplementary Data 5), each with a high consistency score of no less than 0.42, where the consistency score was calculated as the average Spearman correlation coefficient of the cross-brain neurite density distribution for all pairs in that module.

### Tracing local morphology

The local reconstructions were generated using the somas of SEU-S182K. To avoid highly interweaved neurons, somas with more than five neighboring somas within a radius of approximately 128 μm were eliminated. For each soma, a block measuring 512 × 512 × 256 voxels was cropped from the second highest resolution images, with the soma located at the center of the block. This block size corresponds to an approximate diameter of 236 μm in each of the $x$ and $y$ axes, and 512 μm in $z$ axis around the soma, encompassing most of the basal dendrites (98% of total compartments in manual annotations), a portion of the apical dendrite (63%).

We combined two automatic tracing algorithms, All-Path-Pruning (APP2)[84], and the tubular fitting-based algorithm neuTube[85], to trace each image block. Default parameters were used for both algorithms except that the background threshold in APP2 was set to an automatically determined threshold of $\mu + 0.5\sigma$, where $\mu$ and $\sigma$ are the mean and standard deviation of the input image. Reconstructions from APP2 and neuTube were combined to get an initial reconstruction. Specifically, neuTube reconstructions were used to prune the APP2 reconstructions, and nodes in APP2 reconstructions that did not have corresponding nodes within 5 voxels (-2.3 µm) in neuTube reconstructions were pruned. Images were enhanced using a signal-background contrast enhancement method[86] before being subjected to tracing algorithms.

The generated reconstructions were subjected to a segment-pruning pipeline, rectifying possible loops, erratic branches, and intersections with other neurons. Each pruning step operates as an independent filter that takes in the raw neuron tree, and the resulting tree is the intersection of all filtered reconstructions. The detailed pipeline can be summarized as follows:

1. Firstly, a branch pruning was performed to remove any branch that had an excessive angle to its parent ( < 80 degrees) or had an excessive increase in radius (1.5 times the parent branch's radius).

2. Secondly, a crossover pruning step was carried out to expunge branches from putative crossover structures. To do that, we detected all putative crossover structures, including multi-furcating nodes containing more than two child nodes and consecutive bifurcating nodes within five voxels. For each of the crossover structures, we checked all connections between the current branch and its child branches. In specific, branches with small angles (< 80 degrees) were marked as removable. Then, we evaluated branches with mediocre turning angles (80–100 degrees) to confirm if another branch with a large enough angle existed between them (>150 degrees). If such a branch existed, we removed the other branches.

3. Then, a soma pruning strategy was applied to remove branches originating from any other putative somas. A soma candidate was identified when the total area of a candidate node or a set of nearby candidate nodes is large (> 500 pixel² on the maximum intensity projection on the $xy$ plane, corresponding to -105 µm²). A candidate node here is a reconstructed node with a radius larger than five pixels (-2.3 µm) on the $xy$-plane. Nodes within 50 pixels (-23 µm) to the current soma - were not checked. For each detected soma, an integration of deviation angle along the neurite path in between the detected soma and the target soma was estimated, to locate the best cutting position. The deviating angle is the angle between a single local branch and the radial line connecting the soma and the nearer end of the same branch, similar to the G-Cut[43]. We then calculate the integral of deviation angles at both sides, that is, from the current branch to the current soma and the current branch to the putative soma respectively, by weighting the deviation angle with the branch length. Branches with a lower deviation angle integral leading to the putative soma were subsequently removed.

4. Next, a winding pruning was performed to remove any branch that followed a circuitous path to the soma, defined as the ratio between path distance and Euclidean distance being greater than three.

5. Finally, all subsequent nodes of a pruned branch or nodes identified in the previous steps were removed, and reconstructions with fewer than 20 nodes were discarded.

We reconstructed 15,441 local morphologies following this protocol. The morphologies were then mapped to the CCFv3 atlas and automatically labeled the regions of the somas.

## Dendritic microenvironment construction

A dendritic microenvironment was defined as the spatial-tuned fusion of a local morphology and its top five most similar morphologies within a sphere of radius 249 µm. The radius corresponds to the 50th percentile distance among the distances between the 5th closest neuron and the target neuron for all neurons in SEU-D15K. The similarity between two neurons was calculated as the Euclidean distance of their standardized (Z-score normalized) morphological feature vectors.

Each neuron was represented using a 24-dimensional feature vector consisting of 18 L-Measure[Vaa3D] features[87] (except for the "Nodes", "SomaSurface", "AverageDiameter", and "Surfaces" features of the 22 features described in the "Morphology features" section in the Methods), the explained variance ratios of three principal components (variance percentages of PC_1, PC_2, and PC_3), and sum-normalized values of the first principal component (PC_11, PC_12, and PC_13). The principal components (PC_1, PC_2, PC_3) were calculated using principal component analysis (PCA) for all nodes in isotropic space. The variance percentage of a principal component represents the ratio of its variance among that of all principal components. The microenvironment feature is a spatial proximity weighted averaging of features derived from all six neurons constituting the microenvironment. Specifically, the spatial weight for a neuron is the exponential of negatively normalized distance. Therefore, for each microenvironment, its feature was calculated as:

$$F_M = \sum_{i=1}^{6} w_i \times F_i \tag{1}$$

where

$$w_i = \frac{\exp(-d_i/D)}{\sum_{i=1}^{6} \exp(-d_i/D)} \tag{2}$$

where $F_M$ and $F_i$ are the feature vectors of the microenvironment and neuron $i$. $d_i$ is the distance between the soma of the target neuron and its neighboring neuron $i$. Here, $D$ is the sphere radius (249 µm).

To intuitively visualize the data, we used the max-relevant min-redundancy (mRMR) algorithm to reduce the 24-dimensional microenvironment feature vector to the three most discriminating features. The identified top three features were straightness, Hausdorff dimension, and variance percentage of the third principal component (PC_3), which represent the branch bending, the fractal dimension of the morphology, and the explained variance ratio of PC_3, respectively. The automatically selected mRMR features may be replaced by other user-chosen features in the 24-dimensional feature-space. To generate the whole-brain microenvironment map, we initialized an empty image of the same size as the 25 µm resolution CCFv3 atlas and assigned each neuron's three features to the three color-channels (R, G, B) of its soma location in the image. The resulting map contained 15,441 data points that clearly represented the three most discerning microenvironment feature values. Data points located in the right hemisphere were mirrored to the left hemisphere, based on the widely accepted left-right symmetry assumption, which was also validated in this work (Supplementary Fig. S4, Supplementary Fig. S5). Each feature underwent min-max normalization and linearly mapped to a range between 0 and 255 for consistent comparison. Microenvironments located within 1 mm of the middle sections along the axial, sagittal, and coronal views were mapped to the corresponding middle sections using the maximum intensity projection (MIP). The resulting maps were superimposed with the boundary outlines of the CCFv3 atlas to facilitate the semantic analysis of the feature distribution. We employed KMeans clustering to classify regions based on the concatenation of the mean and variance feature vectors of all microenvironments in that region.

### Refining single neuron reconstructions and skeletons

We manually annotated 140 single neurons in their entirety from 19 fMOST mouse brains from the 1U19MH114830-01 project at submicron resolutions. The annotations were carried out with the CAR platform[29]. Together with refined versions of previously released 1736 neurons[16] we aggregated a total of 1876 single-neuron morphologies (SEU-A1876). Each neuron-reconstruction should meet the following criteria: (1) A neuron-reconstruction must have a single soma. (2) A neuron-reconstruction must be a single connected piece (graph). (3) A neuron-reconstruction cannot have any loop. (4) A neuron-reconstruction cannot contain duplicated reconstructed compartments. (5) A neuron-reconstruction can have only bifurcating branches except at the soma. (6) A neuron-reconstruction should have consistently annotated dendritic and axonal segments.

The morphologies annotated at low resolutions frequently lead to image-morphology mismatch in the highest resolution space. To overcome this issue, we developed a retracing strategy to refine the skeleton skewness[45]. This involved a two-step process. First, the skeletons were split into fragments of 50 μm, and every two consecutive middle points of the fragments were connected using the graph-augmented deformable model (GD)[88]. The GD algorithm automatically fits the skeleton to the nearest salient neurite with the constraints of the original skeleton priors. In the second step, an additional step of GD tracing was applied to the middle points of two consecutive refined fragments.

### L-Measure^Vaa3D features

We used L-Measure features to characterize microenvironment, full morphology, and local morphology. We calculated 22 features implemented in the "global_neuron_feature" plugin of Vaa3D: "Nodes", "SomaSurface", "Stems", "Bifurcations", "Branches", "Tips", "OverallWidth", "OverallHeight", "OverallDepth", "AverageDiameter", "Length", "Surface", "Volume", "MaxEuclideanDistance", "MaxPathDistance", "MaxBranchOrder", "AverageContraction", "AverageFragmentation", "AverageParent-daughterRatio", "AverageBifurcationAngleLocal", "AverageBifurcationAngleRemote", "HausdorffDimension".

Note that while the meaning of the L-Measure^Vaa3D features are the similar to these defined in L-Measure server[87] (http://cng.gmu.edu:8080/Lm), some of them slightly differ in implementation. For instance, in Vaa3D, features like "OverallWidth", "OverallHeight", and "OverallDepth" represent the span of all nodes or compartments along the $x$ (anterior-posterior), $y$ (dorsal-ventral), and $z$ (left-right) axes, respectively. However, the 5% extreme values were excluded in L-Measure server.

### Full morphology analysis

We leveraged L-Measure^Vaa3D features (22 dimensions) but excluded five inaccessible features such as soma surface and total surfaces for the manually annotated neurons, resulting in 7 global features and 10 local features for each fully reconstructed neuron. Each of the local features was represented by four statistical characteristics: minimum, maximum, mean, and standard deviation, resulting in a total of a 47-dimensional feature vector. The seven global features are "Stems", "Branches", "OverallWidth", "OverallHeight", "OverallDepth", "OverallVolume", and "Length". The ten local features are "br_length" (path length of a branch), "br_order" (branch order), "br_contraction" (contraction of a branch in L-Measure), "bif_EucDist2soma", "bif_PathDist2soma", "asymmetry" ("Partition Asymmetry" in L-Measure), "ampl_local", "ampl_remote", "tilt_local", and "tilt_remote" (Supplementary Fig. S6). We standardized all the features using Z-score normalization, and evaluated neuronal similarity as the cosine distances between these features. We also considered the spatial relationships between neurons by calculating the Euclidean distance ($d$) of each soma in CCFv3 space. We then computed the exponential of $-d$ for

these two somas after normalization. We defined the similarity of neurons as the product of feature similarity and spatial distance. Spectral clustering was utilized to classify neurons into different clusters. Specifically, we treated the entire dataset as a graph, where the neurons in the dataset served as individual nodes, and the weights of the connections between nodes were defined by the similarities between the neurons.

The silhouette scores between soma locations were estimated with the "metrics.silhouette_score" method of the scikit-learn package, using default parameters, which are calculated as $(b - a) / \max(a, b)$, where ($a$) is the mean intra-cluster distance of a sample, and ($b$) is the distance between the sample and the cluster not containing the sample

### Arbor detection and analysis

An arbor in this work is a dense-packing sub-tree. The dendritic arbors are the complete dendrites. For axons, we utilized spectral clustering to subdivide them into arbors. This was done by creating an undirected graph composed of vertices that represent nodes in the original tree, while the weights of edges between vertex pairs were represented by the exponential of the negative distances between nodes. To facilitate comparison, we utilized the dominant auto-clustering arbor number of neurons in the same brain region using the majority-vote principle. Regional features were calculated by taking the average of the features of all neurons in that region.

We employed the number of branches ("#branch"), total arbor volume ("volume"), and maximal spatial density ("max_density") - the latter being defined as the number of axonal nodes located within a 20 μm radius for every node - as three features to characterize arbor morphology. We differentiated between proximal and distal arbors based on whether or not the Euclidean distance from the maximal density node to the soma exceeded 750 μm. All values of the same feature were min-max normalized to the range of 0 to 1. The arbors of each neuron were sorted by the Euclidean distance-to-soma and designated as "A1", "A2", and "A3" (if present). For the classification of thalamic axonal arbors, we utilized agglomerative clustering by combining 4 features (the 3 aforementioned features and "distance-to-soma") alongside the average projection of the axonal arbors.

### Detect primary axonal tract motifs

The primary axonal tract for a neuron is the longest axonal path without short branches at the terminal side. The process of identifying the primary axonal tract begins by determining the longest axonal path and then iteratively removing all branches shorter than the second-longest axonal branch from the tip of the path towards its soma. The resulting path is the primary axonal tract, and its direction is defined as soma to the terminal.

We then mapped primary tracts to the standardized CCFv3 space, grouping them according to projection subtypes, and estimated the radius profile of each group. To this end, we sub-sampled each tract with 200 uniformly-spaced nodes and calculated the cross-sectional radii of points with corresponding percentiles. In specific, for a given set of points, we computed the distance of every point to their center in the reduced 2-dimensional space generated by principal component analysis (PCA), and extracted the 75th percentile as the radius. Based on the comparison of somas and terminal radii, projection patterns were defined as convergent, divergent, and parallel.

### Detect axonal varicosities

We updated the approach reported previously[77] by combining both intensity and radius profiles along the axonal shafts. We standardized the neuronal images by applying an enhancement pipeline[86] designed to enhance the signal-to-background contrast. The detection process starts with partitioning the axonal skeletons into 20 μm length fragments, along which we calculated the intensity and radius profiles,

leading to the identification of initial candidates for varicosities, which exhibit overlapped peaks in the intensity and radius profiles. False positive results in the initial candidates were filtered out through heuristic criteria that a varicosity should be 1.5 times larger than its surrounding axonal nodes similar to the weight threshold in previous studies[69,89] and have an image intensity value above 120 in 8-bit images (maximum intensity 255). Finally, we remove any possible duplicates by deleting candidates that were closer than five voxels in highest-resolution images, a distance roughly equivalent to the size of a typical varicosity (1–2 μm). All detected varicosities were registered to the CCFv3 atlas along with their morphologies using mBrainAligner. To evaluate the varicosity detection, two independent experts manually annotated 235 image blocks (~59 × 59 × 256 μm³) using CAR-mobile, yielding 1,450 annotated varicosities. This manual annotation dataset was utilized to evaluate the accuracy of the automatic varicosity detection method.

### Cross-scale feature generation

We obtained a 75-dimension cross-scale feature vector (Fig. 1E) for each neuron in SEU-A1876, by concatenating features derived from five distinct morphological scales, including microenvironment, full morphology, arbor, varicosity, and primary axonal tract (motif). The microenvironment features for each neuron were acquired by extracting the microenvironment features of the neuron in the same soma region in SEU-D15K that had the most similar L-Measure[Vaa3D] features. For microenvironment and full morphology, we utilized 18 L-Measure[Vaa3D] features, such as "Bifs", "Branches", and "Tips", as those in "Dendritic microenvironment construction" (Methods). The arbor features consisted of a concatenation of dendritic and axonal arbor features. For comparative reasons, the axons of each neuron were arborized to two arbors. All features were standardized to a normal distribution and concatenated. Features of a region were estimated by averaging all neurons within that region.

We introduced a metric, called the DS matrix, to calibrate the diversity among cell types, encompassing both intra-type and inter-type similarities. Each value (DS value) in the matrix indicates the average correlation coefficient between all neuron pairs for the respective two regions (neuron types). The correlation of two neurons is determined by calculating the Pearson correlation coefficient between their cross-scale features. A higher value in the matrix signifies greater stereotypy, while a lower value signifies greater diversity. The axon-axon "distance" was the Euclidean distance between the projection vectors of the neuron pair, which was released in our previous work (Peng et al.[16]).

### Reporting summary

Further information on research design is available in the Nature Portfolio Reporting Summary linked to this article.

## Data availability

All morphometry data are accessible from both Zenodo (https://doi.org/10.5281/zenodo.13944322) and Google Drive (https://drive.google.com/drive/folders/1NwwTe840_0KQhv-zVLhw58LU9nntkb-F?usp=sharing). Documentation and video demonstrations for the dataset are available at https://sd-jiang.github.io/full_spectrum/. All brain images are generated from projects within the BICCN initiative, with most of them publicly accessible through the Brain Image Library (BIL) at https://www.brainimagelibrary.org/. The remaining 28 brain images will be made available shortly after compilation and uploading. Source data are provided with this paper.

## Code availability

The source codes are available via https://doi.org/10.5281/zenodo.13979929. Dependencies are summarized in the *requirement.txt*, which can be easily installed using the Python package manager, PIP. Detailed documentation and step-by-step instructions are provided in the repository. Video instructions showing how to use the package is available at https://sd-jiang.github.io/full_spectrum/. Vaa3D (version 4.001) and Vaa3D-x (version 1.1.2) are available on the Vaa3D GitHub repository (https://github.com/Vaa3D) in both source code and released binary forms. Every version of Vaa3D encompasses core bindings such as TeraFly and TeraVR, plugins that include "Simple_Adaptive_Thresholding" filter, "global_neuron_feature", Grayscale Image Distance Transform (GSDT), and auto-tracing algorithms like APP2, neuTube, and GD. The updated version of mBrainAligner can be found at https://github.com/Vaa3D/vaa3d_tools/tree/master/hackathon/mBrainAligner. MorphoHub is available at https://github.com/SD-Jiang/MorphoHub. Source codes for varicosity detection can be found at https://github.com/Vaa3D/vaa3d_tools/tree/master/hackathon/shengdian/BoutonDetection. The collaborative augmented reconstruction system (CAR) is available through https://github.com/neurogeom/CAR. The codes for full morphological feature extraction are accessible at https://github.com/Vaa3D/vaa3d_tools/blob/master/hackathon/shengdian/NeuroMorphoLib. Clustering and other machine learning algorithms, including K-Means, spectral clustering, HDBSCAN, and Principal Component Analysis (PCA), come from the third-party package scikit-learn (version 1.2.2) of Python (version 3.10). The Python derivative of mRMR (pymrmr, version 0.1.11, https://github.com/fbrundu/pymrmr) is used in this paper to select the most discriminating features. Hierarchical clustering in module detection and arbor analysis utilizes the *hclust* library from the stats package (version 4.2.2) in R (version 4.2.2).

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

## Acknowledgements

We thank Yong Yao and NIH for coordination of the BICCN effort in data collection and synergized analysis, Brain Image Library (BIL, www.brainimagelibrary.org) for public sharing of the BICCN database, Josh Huang for sharing raw image data, Yuanyuan Song and Lulu Yin for assistance of the single-neuron annotation, Jiangshan Liang, Luchen Deng, Shize Chen, Fei Xing, Yihang Zhu, Lei Huang and Kaixiang Li for help in tool development, Xuan Zhao, Ye Zhong, Jingzhou Yuan, and another 7 anonymous users for help in soma-annotation, and Yiwei Li for assistance of optimizing soma locations. This work was mainly supported by a Southeast University (SEU) initiative of neuroscience and a New Cornerstone grant awarded to H.P.; The Southeast University team was also supported by a MOST (China) Brain Research Project, "Mammalian Whole Brain Mesoscopic Stereotaxic 3D Atlas" (2022ZD0205200 and 2022ZD0205204) awarded to L.L.; L.L. and Y.Liu were also supported by "the Fundamental Research Funds for the Central Universities of China" (No. 2242023K5005); G.A.A. was supported in part by NIH grants R01NS39600 and RF1MH128693. H.Z. was supported by BRAIN Initiative grant U19MH114830. P.M. was supported in part by the Crick-Clay Professorship (Cold Spring Harbor Laboratory) and BRAIN Initiative grant U19MH114821. Q.L. and H.G., were supported by STI2030-Major Projects Grant No. 2021ZD0201001, National Natural Science Foundation of China Grants No. 81827901. Research reported in this publication was also supported by the National Institute of Mental Health under Award Number R01MH131537 to S.D.R. and Z.W., U01MH114824 from National Institute of Mental Health to P.O. and Z.W., the Friedman Brain Institute under a Research Award by the Fascitelli Family to S.D.R. and Z.W., the Mindich Child Health and Development Institute under a pilot grant to S.D.R. and Z.W., and M.G.F. was also supported by the Beatrice and Samuel A. Seaver Foundation and the Fundacio Martin Escudero.

## Author contributions

H.P. conceptualized and managed this study. H.Z., P.O., H.D., Z.W., and S.D.R. provided brain images. M.G.F. prepared several viral labeled brains and W.W. processed the samples and did the LSFM imaging for several brains supplied by Z.W. and S.D.R. S.J. collected and pre-processed brain images. L.L. and X.C. led the single-neuron annotation. L.Z. and Y.H. led the development of tools for soma annotation and neuron reconstruction. S.J. developed the tools for data management, co-developed the skeleton refinement, and detected varicosities. Y.Liu and Z.Z. implemented neurite segmentation. Z.Z. and G.W. reconstructed the local dendrite morphologies. Y.Liu coordinated data analysis and conducted analysis on microenvironments, projection patterns, and cross-scale features, and implemented algorithms for neurite segmentation and arbor feature extraction. S.J. conducted analysis on full morphology and varicosities. Y.Li performed analysis on regional modularization and multi-morphometry visualization. S.Z. and Y.Liu analyzed neuron arbors. Z.Y. conducted the image registration and morphometry mapping to CCF. Yuanyuan L. and L.Q. evaluated the registration robustness. P.Q. assisted with the validation of varicosity detection. Y.Liu and Q.G. reviewed and refactored the source codes. H.G. and Q.L. produced fMOST imaging. H.P. and Y.Liu wrote the manuscript with assistance of all authors, including L.M.G, G.A.A., P.M., M.H., Z.W., S.D.R., who reviewed and revised the manuscript. These authors contributed equally: S.Z., Z.Y., Z.Z.

## Competing interests

The authors declare no competing interests.

## Additional information

[1]New Cornerstone Science Laboratory, SEU-ALLEN Joint Center, Institute for Brain and Intelligence, Southeast University, Nanjing, Jiangsu, China. [2]School of Computer Science and Engineering, Southeast University, Nanjing, Jiangsu, China. [3]School of Biological Science & Medical Engineering, Southeast University, Nanjing, Jiangsu, China. [4]Ministry of Education Key Laboratory of Intelligent Computation and Signal Processing, Information Materials and Intelligent Sensing Laboratory of Anhui Province, School of Electronics and Information Engineering, Anhui University, Hefei, Anhui, China. [5]Seaver Autism Center for Research and Treatment, Icahn School of Medicine at Mount Sinai, New York, NY, USA. [6]Department of Psychiatry, Icahn School of Medicine at Mount Sinai, New York, NY, USA. [7]The Mindich Child Health and Development Institute, Icahn School of Medicine at Mount Sinai, New York, NY, USA. [8]Friedman Brain Institute, Icahn School of Medicine at Mount Sinai, New York, NY, USA. [9]Alper Center for Neural Development and Regeneration, Icahn School of Medicine at Mount Sinai, New York, NY, USA. [10]Appel Alzheimer's Disease Research Institute, Feil Family Brain and Mind Research Institute, Weill Cornell Medicine, New York, NY, USA. [11]Department of Cell, Developmental & Regenerative Biology, Icahn School of Medicine at Mount Sinai, New York, NY, USA. [12]Department of Neuroscience, Icahn School of Medicine at Mount Sinai, New York, NY, USA. [13]Cold Spring Harbor Laboratory, Cold Spring Harbor, NY, USA. [14]HUST-Suzhou Institute for Brainsmatics, JITRI, Suzhou, China. [15]Allen Institute for Brain Science, Seattle, WA, USA. [16]Center for Integrative Connectomics, Department of Neurobiology, David Geffen School of Medicine at UCLA, Los Angeles, CA, USA. [17]State Key Laboratory of Digital Medical Engineering, School of Biomedical Engineering, Hainan University, Haikou, China. [18]Key Laboratory of Biomedical Engineering of Hainan Province, One Health Institute, Hainan University, Haikou, China. [19]Volgenau School of Engineering, George Mason University, Fairfax, VA, USA. [20]These authors contributed equally: Shengdian Jiang, Yingxin Li. ✉e-mail: lijuan-liu@seu.edu.cn; h@braintell.org

