## [Transparent Peer Review file · Nature Communications]

Neuronal Diversity and Stereotypy at Multiple Scales through Whole Brain Morphometry

Corresponding Author: Dr Hanchuan Peng

Version 0:

Reviewer comments:

Reviewer #1

(Remarks to the Author)

My main criticism of the manuscript remains, which is that the work presented (even for a resource) does not show any clear biological findings nor does it provide any biologically relevant explanations for some of the observed statistical differences, so the goal to estimate quantitative differences for some arbitrary parameter in many samples without a clear focus does not fulfil the criteria for a publication of broad interest in neuroscience.

The manuscript continues to be an amalgamation of imaging data with confusing statements and strong conclusions without clear supporting evidence. For example, in the new text the authors write “the boundaries between anatomical regions could be difficult to recognize, such as the border between CP and cortical regions.” – what does this mean? In reality, using standard cytoarchitecture and any type of imaging, the distinction between the cortical regions and the striatum (CP) is probably one of the easiest to identify since the massive myelinated fiber bundle (i.e. the corpus callosum) separated cortex from striatum. Similarly, description of the rationale continues to be confusing throughout the manuscript – as an example the authors write “Based on dendritic microenvironments, one may perform an exhaustive survey of many paths across different 3-D anatomical areas.” – again, I unfortunately cannot understand what this means.

I unfortunately also noticed that the authors did not take the time to properly respond to my main questions, and to a very basic question on the novelty of showing that different neuron types exhibit different dendritic morphologies (e.g. interneuron subtypes vs pyramidal neurons). The supplementary figures presented in response to these questions do not at all answer the questions, but instead just plot more data. The same superficial response was given to the majority of my questions, and I find that overall, the responses were quite superficial and did not really address the main points.

As a general comment, the authors state that they present “one of the largest collections of single-neuron morphology data in mice”, which they call the IMG205 dataset to indicate that it consists of 205 mouse brains: in my understanding the IMG205 dataset does not actually include imaging information from 205 brains, even if this number is again stated throughout the manuscript to seemingly inflate the effort. The reason for this unclear description of the underlying data is unclear but reinforces the impression that the authors exaggerate their effort to impress the reader.

What became clear from the information in Suppl. Table 1 is that almost all the imaging data that is presented in this manuscript is already published by the corresponding author (Peng, et al, Nature 2021). The authors make no effort in explaining why they have analyzed this dataset, the criteria for selecting their parameters, and how their findings can explain or predict any biological function. According to the description in Suppl. Table 1, the full neuron morphology has been quantified in only a small sample of the brains, and there is no systematic approach to what samples have been used for full neuron reconstruction or dendrite morphology reconstruction, and no rationale for how selection was made for the quantification of samples with labeling of different neuron types (e.g. interneuron subtypes).

In summary, this resource could be of value to the field if it was analyzed and presented in a structured and logical way, since it is ambitious in scope, but does not answer or question any neuroanatomical or circuit questions that the field can use for future investigations.

Reviewer #2

(Remarks to the Author)

The work presented by Liu et al. provides a very valuable resource in the field of whole-brain morphometry.

I reviewed the manuscript during its submissions to another journal. The authors have thoroughly addressed my previous concerns. I have no further issues to raise.

The manuscript is well-polished and ready for publication in its current form.

Point-by-point response to NCOMMS-24-35238-T: “Full-Spectrum Neuronal Diversity and Stereotypy through Whole Brain Morphometry”

Comments from Reviewer #1

Overall comment 1: *My main criticism of the manuscript remains, which is that the work presented (even for a resource) does not show any clear biological findings nor does it provide any biologically relevant explanations for some of the observed statistical differences, so the goal to estimate quantitative differences for some arbitrary parameter in many samples without a clear focus does not fulfil the criteria for a publication of broad interest in neuroscience.*

Response: We respectfully disagree with the criticism of lack of biological findings. There should be misunderstanding of the significance of our work.

To clarify, we have summarized the key findings in a table (see **Figure 1** below; **Supplementary Table S8** in the manuscript). Indeed, our findings can be seen at two levels.

Firstly, we quantitatively identified morphological patterns at six different scales covering nearly commonly analyzed morphological levels (see **Figure 2** below):

1. We identified 18 modules from the 314 CCFv3 brain regions through population-level neurite analyses of 191 submicron resolution whole mouse brains.
2. We found that using an ensembled representation of dendritic morphologies (“microenvironment”) help identify sub-regional spatial organization while aligns well with the CCF atlas overall.
3. We categorized single-neuron full morphologies into four distinct clusters using our spatial-enhancement techniques.
4. We observed stereotyped arborization patterns in cortical, striatal, and thalamic neurons.
5. We confirmed quantitatively the convergent projection patterns for striatal neurons and divergent patterns for thalamic and cortical neurons using a simplified yet unbiased representation: primary axonal tracts.
6. We identified stereotyped axonal varicosity distributions across different brain areas and also their branch-level preferences.

Secondly, we proposed a cross-scale integrative analysis for the first time and demonstrated that morphometry at different scales complements the classification of neuron types. Together, they revealed a highly modularized neuroanatomical organization with respect to brain regions, cortical laminations, and projection types of cortical neurons.

To our knowledge, these patterns are unprecedented in previous studies. Therefore, the claim that our work “*does not show any clear biological findings nor does it provide any biologically relevant explanations for some of the observed statistical differences*” is incorrect.

However, if you can provide specific examples where our results had been previously reported by others, we would be happy to cite those sources and revise the paper accordingly.

Level	Morphology type	Scale	Morpho-features	Whole-brain registration needed?	Data	Previous knowledge	Previous major literatures	Our key findings	Major location of our results in Figures / Tables	Our data sharable?	Shared results data volume / size
Whole brain	Cross-scale	1µm - 10000+ µm	Multiscale features from neuron population to axonal varicosity	Yes	Petavoxels	NO existing systematic cross-scale study	NO	1. Highly-modularized neuroanatomical organizations are observed in the central nervous system (CNS) neurons, with respect to brain regions, cortical laminations, and projection types of cortical neurons. 2. Morphometry at different scales complements the classification of neurons in the central nervous system.	Figure 1; Figure 8	Yes	3.7 Petavoxels
Neuron population (global)	Neuron population	1000µm-10000µm	Number of neurite voxels across all brain regions	Yes	Petavoxels	Some whole-brain modularity of mammalian brains have been analyzed at macro- or meso-scales	1. Bertolero et al., PNAS, 2015, doi: 10.1073/pnas.1510619112 2. Wang et al., J Neurosci, 2012, doi: 10.1523/JNEUROSCI.6063-11.2012 3. Oh et al., Nature, 2014, doi: 10.1038/nature13186	We identified 18 new modules from the complete set of 314 brain regions at a submicron resolution.	Figure 20; Supplementary Table S5	Yes	3.7 Petavoxels
Neuron population (local)	Dendritic microenvironment	30 million µm ³	Spatially enhanced morphological features	Yes	Thousands to millions microenvironments	Parcellations reported in low resolution, incapable of differentiating subregions that containing neurons with different morphology and function.	1. Lotin et al., Nature, 2007, doi: 10.1038/nature05453 2. Oh et al., Nature, 2014, doi: 10.1038/nature13186 3. Dong et al., Wiley, 2008 4. Paskos et al., Elsevier, 2001, 2012 5. Wang et al., 2020, doi: 10.1016/j.cell.2020.04.007	Microenvironment representation of neighboring neurons discriminates sub-regional parcellation	Figure 3	Yes	15441
Single-neuron	Full morphology	500µm-5000µm	Spatially enhanced Morphological features	Yes	Hundreds to thousands neurons	Single neurons could not be classified into discrete clusters, based on conventional features of projection morphology.	1. Peng et al., Nature, 2021, doi: 10.1038/441586-021-03941-1 or 2. Wilmshast et al., Cell, 2019, doi: 10.1016/j.cell.2019.07.042	We identified 4 discrete cross-areal single neuron clusters from 1,876 single neurons, covering 94 brain regions. This is the first time such clear clustering has been shown, and is only possible with our enhanced analysis	Figure 4	Yes	1876
Sub-neuronal	Sub-neuronal arbor	100µm-2000µm	Morphological features (e.g., volume and path length)	No for sub-areas, yes for comparative studies	Thousands arbors	NO previous study	NO	Neuronal arborization pattern characterizes cortical, striatal, and thalamic neurons in mouse.	Figure 5	Yes	3776
	Primary axonal tracts	100µm-2000µm	Volume, length, and coordinates	Yes	Hundreds to thousands tracts	Projection patterns were experimentally estimated through various labeling techniques or transcriptome profiling, however quantifying the relationship between sources (somata) and targets (terminal axons) has been challenging due to their imbalanced scales.	1. Li et al., Brain Structure and Function, 2021, doi:10.1007/s00429-021-02289-6; Oh et al., Nature, 2014, doi: 10.1038/nature13186 2. Fernández-Nogales, Advanced Science, 2023, doi:10.1002/adv.202200615 3. Wang et al., Neurosci Bull, 2021, doi: 10.1007/s12264-020-00616-1 4. Peng et al., Nature, 2021, doi: 10.1038/441586-021-03941-1	Confirmed quantitatively the convergent projection patterns for CP neurons, and divergent patterns for cortical and thalamic neurons.	Figure 6	Yes	1876
	Axonal varicosity	1µm-3µm	Customized features (see Figure 1E)	No for sub-areas, yes for comparative studies	Millions [predicted] boutons	Comparative analyses of the spatial preference of varicosities across the whole brain have not been found, and the distribution of local varicosities was evaluated only in small datasets, never at the whole brain scale	1. Julian et al., PLoS Computational Biology, 2010, doi:10.1371/journal.pcbi.1000711 2. Karube et al., J Neurosci, 2004, doi: 10.1523/JNEUROSCI.4814-03.2004	1. Stereotyped axonal varicosity features are observed among neurons from the same brain structures (e.g., cortex, thalamus, and cerebral nuclei), while diverse across different structures. 2. Axonal branches successive to a varicosity-containing branch (near the soma) always contain varicosities	Figure 7	Yes	3.63 millions

Red color: key novelties. See the main text and Supplementary Figures / Tables for details. See video demos etc at https://sd-jiang.github.io/full_spectrum/

Figure 1. Summary of key findings

Figure 2. Common morphological scales

Comment 2: The manuscript continues to be an amalgamation of imaging data with confusing statements and strong conclusions without clear supporting evidence. For example, in the new text the authors write “the boundaries between anatomical regions could be difficult to recognize, such as the border between CP and cortical regions.” – what does this mean? In reality, using standard cytoarchitecture and any type of imaging, the distinction between the cortical regions and the striatum (CP) is probably one of the easiest to identify since the massive myelinated fiber bundle (i.e. the corpus callosum) separated cortex from striatum. Similarly, description of the rationale continues to be confusing throughout the manuscript – as an example the authors write “Based on dendritic microenvironments, one may perform an exhaustive survey of many paths across different 3-D anatomical areas.” – again, I unfortunately cannot understand what this means.

Response: Thank you for bringing these concerns to our attention. We believe there was a misunderstanding.

Please note that this paper is not about imaging data. Instead, it is about the extracted patterns from the brains at various spatial and anatomical scales.

For the comment “... *in the new text the authors write “the boundaries between anatomical regions could be difficult to recognize, such as the border between CP and cortical regions...”*, it seems you had extracted the sentence out of its context. Our comparison was intended to highlight the differences in microenvironment features relative to the averaged brain data from hundreds of Nissl-stained images used in the CCF atlas generation. To avoid confusion, we have removed the description in the main text (lines 338-340).

Regarding the comment “...*Based on dendritic microenvironments, one may perform an exhaustive survey of many paths across different 3-D anatomical areas...*”, we rewrote as “The whole-brain dendritic microenvironments could facilitate the exploration of both inter-regional and intra-regional organization across various brain areas, in addition to the four exemplar paths.” (lines 352-353).

Comment 3: *I unfortunately also noticed that the authors did not take the time to properly respond to my main questions, and to a very basic question on the novelty of showing that different neuron types exhibit different dendritic morphologies (e.g. interneuron subtypes vs pyramidal neurons). The supplementary figures presented in response to these questions do not at all answer the questions, but instead just plot more data. The same superficial response was given to the majority of my questions, and I find that overall, the responses were quite superficial and did not really address the main points.*

Response: We thank you for acknowledged that we had added more data. We respectfully disagree with the assertion that our responses were superficial and “*do not at all answer the questions*”. We have thoroughly revised the manuscript and provided the requested materials in the last revision. If there are specific missing materials, we would appreciate it if you could specify them so that we can address them accordingly.

Since this comment in this review report has little detail, we would like to elaborate our revision as follows, based on the previous review report. We hope this effort can help clarify our manuscript.

- **Previous comment 2:** We have already rephrased the expression the reviewer mentioned for clarification in the revised manuscript (lines 239-240). The term “connections” here refers to the high correlation between regions based on their neurite co-occurrence across 191 brains, indicating possible projectional or functional connections between these regions. Thus, the mentioned sentence means that our extracted “connections” align well with existing anatomical studies (e.g., CCFv3) and single neuron morphologies (Peng et al., Nature, 2021, doi: 10.1038/s41586-021-03941-1).

- Previous comment 3:** The reviewer stated, *“it is unclear what this metric and quantification actually captures”*, we have clarified in the manuscript that the “neurite density” refers to the total number of detected neurite voxels within a specific region, divided by the total number of neurite voxels detected in the entire brain (lines 1364-1365). Biologically, the neurite density vector for each brain may suggest possible projection pathways for sparsely labeled neurons, though it is not a one-to-one correspondence. To address this, we aggregate densities for 191 brains to create a composite matrix, wherein each region has a 191-element vector representing its neurite pattern. This approach allows us to estimate the potential co-existence on projection or function pathway based on neurite distribution patterns between region pairs across 191 brains. Regarding *“show segmented images from the imaged neurons that demonstrate the different types of signals they use for their downstream analysis”*, several exemplar image blocks are shown in **Supplementary Figure S3A** and we supplemented more examples in **Supplementary Figure S11**.
- Previous comment 4:** The reviewer would like materials to *“show visualization of key examples from neurites and the region population averages for some of the highly correlated regions (e.g. from the 16 modules)”*. For the first part of suggestion (*“key examples from neurite”*), we have provided examples demonstrating the actual neurite connection between striatum and globus pallidus (**Supplementary Figure S11**). For the second part (*“the region population averages”*), this is what we estimate the correlation and “connection”, and we can definitely provide such information.
- Previous comment 6:** It is important to clarify that the novelty of our work on the local dendritic microenvironment (**Figure 3**) is not *“showing that different neuron types exhibit different dendritic morphologies”*. Instead, the key finding of this specific piece of work is that we found ensembled local dendritic morphologies have a similar spatial layout with the commonly used CCF atlas, while providing better discrimination of anatomical regions compared to the CCF atlas (**Figure 3; Supplementary Figure S16**). We have addressed this clarification in our response to Comment 6 and have also added a table (**Supplementary Table S8; Figure 1** in this response letter) to underscore the key novelties of our research, including the microenvironment. Furthermore, we cannot agree with the assertion that *“Fig. 3 most likely accurately captures some basic aspects of dendrite morphology that is known to exist between neuron types, for example excitatory cortical neurons in different layers or GABAergic neurons in the basal ganglia”*. Instead, according to cutting-edge whole-brain cell atlas research (Zhang et al., Nature, 2023, doi: 10.1038/s41586-023-06808-9; Yao et al., Nature, 2023, doi: 10.1038/s41586-023-06812-z), cortical neurons exhibit co-expression of GABAergic and glutamatergic neurons. Therefore, our microenvironment representation does reflect the neuroanatomical organization of heterogeneous neuron types, but not the *“the basic aspects of dendrite morphology that is known”* between excitatory cortical neurons in different layers. If the reviewer wishes to see the morphologies of different neuron types, we have provided several examples in **Supplementary Figures S15 and S16**. However, it is important to note that this is not the primary focus of our paper.

Comment 4: *As a general comment, the authors state that they present “one of the largest collections of single-neuron morphology data in mice”, which they call the IMG205 dataset to indicate that it consists of 205 mouse brains: in my understanding the IMG205 dataset does not actually include imaging*

information from 205 brains, even if this number is again stated throughout the manuscript to seemingly inflate the effort. The reason for this unclear description of the underlying data is unclear but reinforces the impression that the authors exaggerate their effort to impress the reader. What became clear from the information in Suppl. Table 1 is that almost all the imaging data that is presented in this manuscript is already published by the corresponding author (Peng, et al, Nature 2021). The authors make no effort in explaining why they have analyzed this dataset, the criteria for selecting their parameters, and how their findings can explain or predict any biological function. According to the description in Suppl. Table 1, the full neuron morphology has been quantified in only a small sample of the brains, and there is no systematic approach to what samples have been used for full neuron reconstruction or dendrite morphology reconstruction, and no rationale for how selection was made for the quantification of samples with labeling of different neuron types (e.g. interneuron subtypes).

Response: There should be a misunderstanding regarding the data we have collected for this study.

The focus of this study is indeed not about the imaging data, but the neuronal patterns extracted from the brain images at different spatial and anatomical scales.

We have provided the imaging dataset IMG204 that consists of 204 mouse brain images. Only 53 were previously “published” in our earlier paper (Peng et al., Nature, 2021, doi: 10.1038/s41586-021-03941-1) (please note that the cited paper did not focus on the imaging data either), while the majority of IMG204 are released for the first time. All images were spatially registered to CCFv3, along with neurite density estimation (Figure 2), so the claim “in my understanding the IMG205 dataset does not actually include imaging information from 205 brains, even if this number is again stated throughout the manuscript to seemingly inflate the effort” is incorrect.

We have also provided 1,876 single-neurons’ full reconstructions. Every reconstruction is a new version that was optimized especially to matching the neuronal skeletons to the respective imaging data, which was nontrivial. This entire new dataset was not published before.

Importantly, in this study we have released several other morphometry datasets, which include 2.6 million putative varicosities, 3776 dendritic and axonal arbors, 182,497 annotated somas, and 15,441 auto-traced local morphologies. These resources are among the largest datasets currently available, not published before.

There is a systematic methodology behind which samples have been utilized for full neuron reconstruction or dendrite morphology analysis. In fact, we auto-reconstructed all potentially annotated somas ($n=182,497$) and subsequently conducted a morphological check to filter out the majority. The reconstruction of full neuron morphology is based on our quality evaluation conducted by experts.

Regarding the labeling strategies and neuron types, we have made extensive efforts to include a substantial dataset comprising 3.7 petavoxels of data and 204 whole mouse brains. These brain images were collected from five different research groups, covering three different modalities and 34 transgenic lines. This constitutes one of the largest and most diverse morphometrical datasets available. The neurons were reconstructed based on their quality, as judged by anatomical experts. For the auto-tracing of

dendritic morphologies, we traced all brains, reconstructing approximately 180,000 morphologies, and then filtered most of them through a quality control process. Therefore, the claim that "*the full neuron morphology has been quantified in only a small sample of the brains, and there is no systematic approach to what samples have been used for full neuron reconstruction or dendrite morphology reconstruction*" is inaccurate.

Comment 4: *In summary, this resource could be of value to the field if it was analyzed and presented in a structured and logical way, since it is ambitious in scope, but does not answer or question any neuroanatomical or circuit questions that the field can use for future investigations.*

Response: It is unfortunate there was a misunderstanding of the substantial amount of novelty documented in this paper. If you could provide specific examples of content from this paper that is not novel and has been published before, we would be happy to cite those sources and make any necessary revisions.

To our knowledge, this is the first holistic multi-morphometry analysis that integrates morphometrics from six spatial scales, ranging from neuron populations to putative synaptic structures. This innovative approach and the study's multi-morphometric nature is a logical organization. We would greatly appreciate it if you could specify which parts of the manuscript you find unstructured.

Comments from Reviewer #2

Overall comment: *The work presented by Liu et al. provides a very valuable resource in the field of whole-brain morphometry.*

I reviewed the manuscript during its submissions to another journal. The authors have thoroughly addressed my previous concerns. I have no further issues to raise.

The manuscript is well-polished and ready for publication in its current form.

Response: We greatly appreciate your recognition of the significance of our work. We also thank you for the suggestions provided during its submission to another journal. Your feedback has been instrumental in refining and improving the manuscript.